# Three-dimensional growth reveals fine-tuning of 5-lipoxygenase by proliferative pathways in cancer

Tamara Göbel[1], Bjarne Goebel[1], Marius Hyprath[1], Ira Lamminger[1], Hannah Weisser[1], Carlo Angioni[2], Marius Mathes[1], Dominique Thomas[2,3], Astrid S Kahnt[1]

The leukotriene (LT) pathway is positively correlated with the progression of solid malignancies, but the factors that control the expression of 5-lipoxygenase (5-LO), the central enzyme in LT biosynthesis, in tumors are poorly understood. Here, we report that 5-LO along with other members of the LT pathway is up-regulated in multicellular colon tumor spheroids. This up-regulation was inversely correlated with cell proliferation and activation of PI3K/mTORC-2– and MEK-1/ERK-dependent pathways. Furthermore, we found that *E2F1* and its target gene *MYBL2* were involved in the repression of 5-LO during cell proliferation. Importantly, we found that this PI3K/mTORC-2– and MEK-1/ERK-dependent suppression of 5-LO is also existent in tumor cells from other origins, suggesting that this mechanism is widely applicable to other tumor entities. Our data show that tumor cells fine-tune 5-LO and LT biosynthesis in response to environmental changes repressing the enzyme during proliferation while making use of the enzyme under cell stress conditions, implying that tumor-derived 5-LO plays a role in the manipulation of the tumor stroma to quickly restore cell proliferation.

## Introduction

Although the expression of arachidonate 5-lipoxygenase (5-LO), the central enzyme in leukotriene (LT) biosynthesis, is restricted to certain leukocyte subtypes in healthy individuals (Borgeat & Samuelsson, 1979; Poubelle et al, 1987; Spanbroek et al, 2000), the enzyme has been found frequently expressed in solid cancers of various origins, among them malignant tissues of the gastrointestinal tract, and the reproductive and the central nervous system (Gupta et al, 2001; Hennig et al, 2002; Nielsen et al, 2003; Matsuyama et al, 2004; Wang et al, 2015; Xingfu et al, 2015). This non-physiological 5-LO expression is positively correlated with tumor growth, microvessel density, and metastasis, thus worsening the

prognosis of the tumor (Jiang et al, 2003; Barresi et al, 2007; Wasilewicz et al, 2010; Wang et al, 2015; Bai et al, 2018).

A number of in vitro experiments could show that the knockdown (KD) of 5-LO affects the growth and survival of cancer cell lines derived from solid tumors (Romano et al, 2001; Tong et al, 2002; Ding et al, 2012; Woo et al, 2017; Monga et al, 2020; Li et al, 2021) and knockout of 5-LO influences the expression of proteins involved in cellular adhesion, extracellular matrix formation, and cytoskeleton organization, thereby affecting directed cell motility and tumor spheroid formation of cancer cell lines (Weisser et al, 2022). Furthermore, 5-LO products are known to activate anti-apoptotic signaling pathways and promote tumor cell proliferation in vitro and metastasis and angiogenesis in vivo (Boado et al, 1992; Hong et al, 1999; Avis et al, 2001; Romano et al, 2001; Chen et al, 2006). Curiously, LT formation in vitro is only low in 5-LO–expressing tumor cells and the factors that trigger 5-LO expression and control the enzyme's activity in solid malignant tissues are as yet only poorly understood. (Weisser et al, 2022).

5-LO is a dioxygenase that inserts molecular oxygen into the membrane-derived PUFA arachidonic acid (ARA) resulting in the formation of $LTA_4$, which is then further converted to $LTB_4$ and the cysteinyl LTs by $LTA_4$ hydrolase ($LTA_4H$) and $LTC_4$ synthase ($LTC_4S$), respectively (Peters-Golden & Henderson, 2007). As reactive oxygen species accumulate during the enzymatic reaction, active 5-LO represents a threat to healthy cells (Harrison & Murphy, 1995; Cho et al, 2011) because 5-LO products such as $LTA_4$ and fatty acid hydroperoxides can form DNA adducts (Hankin et al, 2003; Zhu et al, 2006) and signal ferroptosis (Liu et al, 2015; Lee et al, 2022; Wu et al, 2022). Accordingly, highly active 5-LO is only expressed in short-lived immune cells such as granulocytes. It is therefore not surprising that the enzyme's activity is tightly suppressed in conventional two-dimensional (2D) cell culture of 5-LO–overexpressing tumor cells.

It is well known that the 2D cell culture of cancer cells does not adequately represent the in vivo situation of a tumor where nutrient and oxygen supply is scarce and toxic waste products accumulate. In contrast, cancer cell lines grown as multicellular tumor spheroids (MCTS) in vitro mimic the in vivo situation of a solid malignancy more closely (Weiswald et al, 2015). These spheroids

---

[1]Institute of Pharmaceutical Chemistry, Goethe University, Frankfurt, Germany   [2]Institute of Clinical Pharmacology, Pharmazentrum Frankfurt, ZAFES, Goethe University, Frankfurt, Germany   [3]Fraunhofer Institute of Translational Medicine and Pharmacology ITMP, Frankfurt, Germany

Correspondence: kahnt@pharmchem.uni-frankfurt.de

quickly develop a gradient of small soluble molecules such as nutrients and growth factors, and oxygen and catabolites from surface to the spheroid core, resulting in nutrient deprivation, waste accumulation, restricted oxygen supply, and acidification in the center of cellular structures larger than 500 μm (Lin et al, 2008). Accordingly, MCTS consist of an apoptotic/necrotic core surrounded by a quiescent viable mantle of cells encased by a proliferating outer rim. Furthermore, 3D cell culture mimics the cell-to-cell contact and extracellular matrix formation present in tumor lesions in vivo.

In the present study, we were interested in whether 3D cell culture, which resembles the in vivo situation of the tumor cells more closely, influences the LT formation and expression of the enzymes involved in the LT biosynthetic cascade. For these experiments, the two colon carcinoma cell lines HT-29 and HCT-116 showing robust 5-LO expression in 2D cell culture were chosen.

# Results

### Three-dimensional growth as MCTS influences the expression of enzymes involved in the LT cascade in HT-29 and HCT-116 cells

We have recently shown that LT formation is low in HT-29 and HCT-116 cells although all enzymes important for biosynthesis of these lipid mediators are expressed (Weisser et al, 2022). Because 2D cell culture does not adequately mirror the in vivo situation of a tumor cell, we aimed to investigate whether expression and activity of the LT cascade enzymes are influenced when HT-29 and HCT-116 cells are grown as MCTS. Spheroid formation was induced by seeding the cells in low attachment microplates (50,000 cells/well; 96-well format) for 4 and 7 d without medium change. Considering the higher proliferation rate of cells grown in conventional cell culture, identical cell numbers per well were used for the monolayer controls, but the cells were grown in 12-well microplates instead. Both cell lines formed dense MCTS already after 4 d of culture, which grew wider with time (Fig 1A). HT-29 cells formed smaller MCTS compared with HCT-116 cells (Fig 1B). The monolayer controls of both cell lines were still subconfluent on day 4 while visibly overgrown after 7 d of culture (Fig 1A).

Then, the expression of the enzymes involved in the LT biosynthetic cascade (cytosolic phospholipase $A_{2\alpha}$ [$cPLA_{2\alpha}$], 5-LO, 5-LO–activating protein [FLAP], $LTA_4H$, and $LTC_4S$) was investigated in MCTS and monolayer controls and compared with cells grown under optimal cell culture conditions (co; maximum confluency 70–80%, daily medium change). The qRT-PCR and immunoblot analysis revealed that 3D cell culture leads to profound changes in the expression of several enzymes involved in LT formation (Fig 1D and E). *ALOX5* mRNA and protein were significantly up-regulated in MCTS of both cell lines on days 4 and 7. The same expression pattern was also observed for $LTA_4H$ and $LTC_4S$ in HT-29 cells. In HCT-116 cells, $LTA_4H$ was also highly up-regulated in the MCTS, whereas *LTC4S* mRNA was only slightly but not significantly elevated. In contrast to this, $cPLA_{2\alpha}$ expression was down-regulated upon spheroid formation. The expression of FLAP was not influenced by 3D growth. We also confirmed these expression patterns in MCTS generated with the liquid overlay technique (Fig S1). Of note, the expression of 5-LO

was also up-regulated in densely grown monolayers of both cell lines after 7 d. Cells grown in a 3D environment proliferate less, and a large proportion of the cells within the spheroid are quiescent (Jackson, 1989). Because 5-LO expression was also induced in densely grown monolayers, we performed a cell cycle analysis of these cells. Indeed, these experiments show that dense growth triggered a G0/G1 arrest after 7 d in both cell lines. The proportion of quiescent cells was higher in HT-29 cells (Fig 1C).

Our data revealed that HCT-116 and HT-29 cells grown under optimal cell culture conditions suppress 5-LO but up-regulate the enzyme when grown in a 3D environment. Therefore, we were interested in whether 5-LO activity is up-regulated along with enzyme expression in the MCTS. Because $cPLA_{2\alpha}$ expression was low, all incubations received exogenous ARA. Comparison of intact MCTS with monolayer-grown cells led to significantly reduced 5-HETE formation in HT-29 and HCT-116 cells, presumably because of reduced ARA diffusion into the spheroid core during the assay (Fig 1F). Interestingly, $LTB_4$ was not found in these incubations. To get a more precise picture of the total 5-LO activity in spheroid-grown cells, spheroids were digested and the assay was repeated with the resulting cell suspension to guarantee optimal substrate availability. As expected, 5-HETE formation was elevated to control levels in the digested MCTS of HT-29 cells when substrate availability was high. In contrast, digested HCT-116 spheroids showed no elevation in 5-LO activity. Again, $LTB_4$ was not found in any incubation (Fig 1F).

PG are a group of pro-inflammatory eicosanoids with pro-tumorigenic functions (Jara-Gutiérrez & Baladrón, 2021). Cyclo-oxygenases (COX), the central enzymes in PG formation, can compete with 5-LO for intracellular ARA. We were therefore interested in whether 3D cell culture also influences the expression of the COX isoenzymes in HT-29 and HCT-116 cells and thus PG formation. The mRNA expression of *PTGS1* (COX-1) followed a pattern comparable to *ALOX5*: it was up-regulated in MCTS and dense monolayers after 7 d. *PTGS2* (COX-2) mRNA was only elevated in HT-29 spheroids (Fig S2). There was no COX-2 expression in HCT-116 cells on the protein level, and COX-1 expression was barely detectable showing no regulation upon 3D growth (Fig S2). In HT-29 cells, both COX isoenzymes were detected on the protein level although expression was rather low. Again, no regulation was found upon 3D growth. In line with these findings, no PG release was detectable in the cell culture supernatants of the monolayer and MCTS even after ARA supplementation.

In addition to LT formation, a number of chemo- and cytokines were analyzed in the cell culture supernatants of HT-29 and HCT-116 MCTS after 7 d (Fig 1G). Compared with monolayer-grown cells, VEGF release from MCTS significantly increased in both cell lines by about 3.4-fold (HT-29) and 5.8-fold (HCT-116). Also, interleukin-8 (IL-8) levels were significantly higher in the MCTS supernatants (HT-29, 2.9-fold; HCT-116, 44.4-fold). In contrast, tumor growth factor β1 (TGF-β1) release was potently attenuated in both cell lines when grown as MCTS (HT-29, sixfold; HCT-116, 10.3-fold).

### 5-LO expression is regulated by PI3K/mTOR and MEK-1/ERK signaling in HT-29 and HCT-116 cells

It was recently shown that 3D growth of HT-29 and HCT-116 cells attenuates AKT/mTOR/S6K and MEK-1/ERK signaling (Riedl et al,

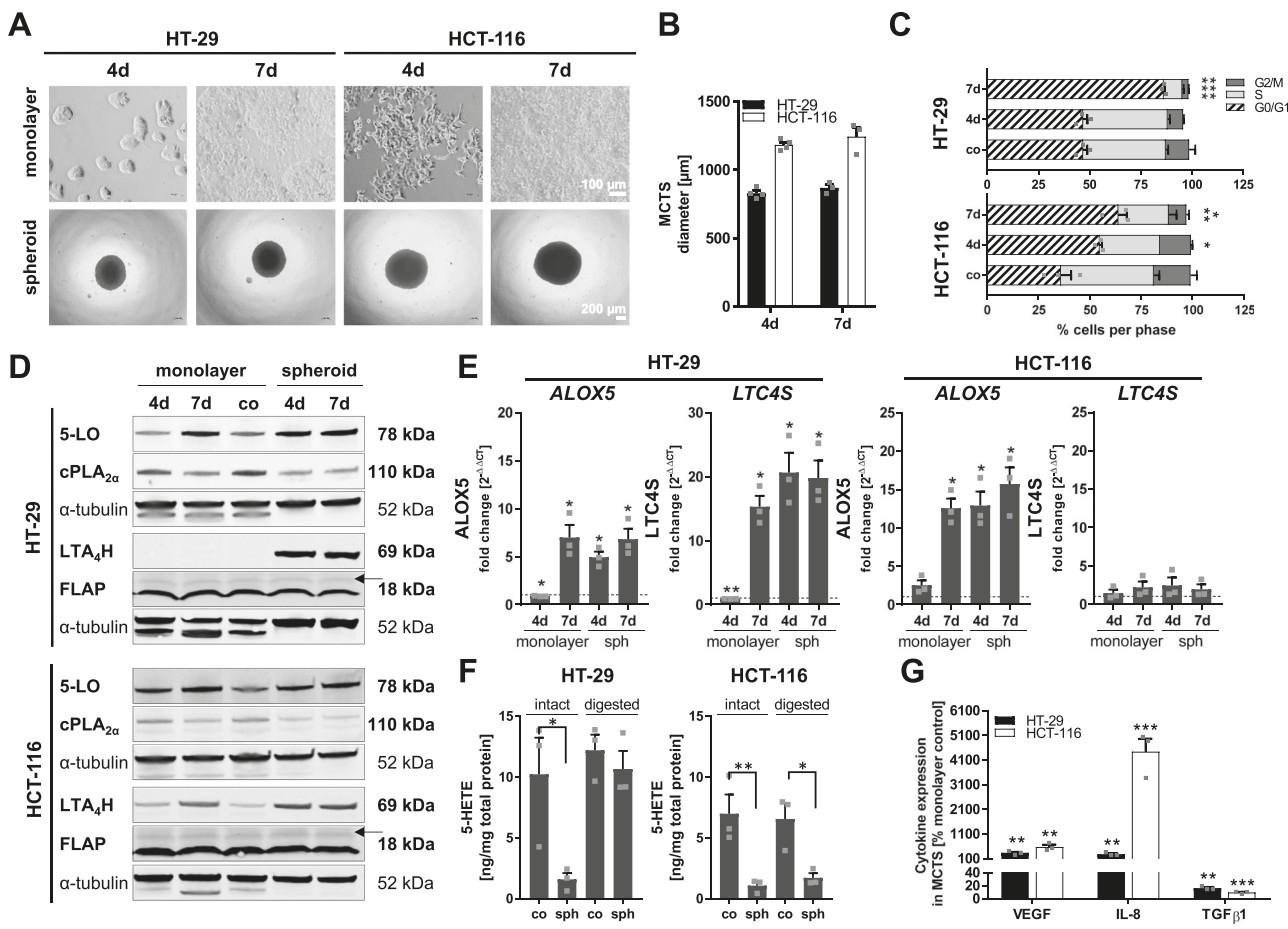

**Figure 1. Three-dimensional cell culture potently influences the expression of enzymes involved in the LT cascade in HT-29 and HCT-116 cells.**
**(A)** Representative pictures of the cell monolayers and MCTS grown for 4 or 7 d (seeding density $0.05 \times 10^6$ cells per well, respectively). Pictures were taken employing a light microscope (monolayers, scale bar: 100 $\mu$m; MCTS, scale bar: 200 $\mu$m). One representative picture of three is shown. **(B)** Average diameters of MCTS grown for 4 or 7 d (4 d, n = 4; 7 d, n = 3). **(C)** Cell cycle distribution of cells grown as monolayers for 4 or 7 d compared with monolayer controls (co), which were harvested before confluency after 48 h of culture receiving daily medium changes. Cells were analyzed via flow cytometry. **(D)** Protein expression of 5-LO, cPLA$_{2\alpha}$, LTA$_4$H, and FLAP. The growth medium in the subconfluent monolayer controls (co) was changed daily, and cells were harvested after 48 h of culture to prevent cell stress. One representative Western blot of three is shown. **(E)** mRNA expression of 5-LO (*ALOX5*) and LTC$_4$ synthase (*LTC4S*). Expression was normalized to the housekeeping gene *ACTB* and the respective monolayer control ($2^{-\Delta\Delta CT}$ method). **(F)** 5-HETE release from intact and digested HT-29 and HCT-116 MCTS and respective monolayer controls. Intact and digested MCTS, and monolayer controls were stimulated with 2.5 $\mu$M Ca$^{2+}$ ionophore (A23187) in the presence of 1 mM CaCl$_2$ (10 min, 37°C). All incubations received exogenous ARA (20 $\mu$M). 5-HETE formation was determined by LC-MS/MS analysis and normalized to the total protein content for each sample. **(G)** VEGF, IL-8, and TGF-$\beta$1 release from MCTS in relation to the respective monolayer (in %) grown for 7 d. Mediators were determined via Cytometric Bead Array (CBA; VEGF, IL-8) or ELISA (TGF-$\beta$1). Mean release in cell monolayers: HT-29 (VEGF, 5,388 pg/10$^6$ cells; IL-8, 272 pg/10$^6$ cells; and TGF-$\beta$1, 51,222 pg/10$^6$ cells); and HCT-116 (VEGF, 3,830 pg/10$^6$ cells; IL-8, 50 pg/10$^6$ cells; and TGF-$\beta$1, 149,397 pg/10$^6$ cells). Data information: results shown in (B, C, E, F, G) are depicted as the mean + SEM from three independent experiments if not stated otherwise. Squares indicate single data points. **(C, E, F, G)** Asterisks indicate significant changes versus co determined by two-way ANOVA coupled with Dunnett's post-test for multiple comparisons (C) or by a *t* test with Welch's correction (E, F, G). *$P < 0.05$, **$P < 0.01$, and ***$P < 0.001$. d, days; co, subconfluent monolayer control; MCTS, multicellular tumor spheroids.

2017). Therefore, we hypothesized that treatment of HT-29 and HCT-116 monolayers with inhibitors of both signaling cascades should partially mimic spheroid growth leading to an up-regulation of 5-LO protein. Before inhibitor treatment, HT-29 and HCT-116 monolayers underwent a short cell cycle synchronization step by serum starvation for 24 h, which did not affect 5-LO expression (Fig S3). Furthermore, it was assured that the inhibitor concentrations used were not cytotoxic to the cells (Fig S4). Then, the cells were treated with several inhibitors targeting the PI3K/mTOR and MEK-1/ERK signaling cascades for a further 24 h. Subsequently, *ALOX5* mRNA expression and 5-LO protein expression were assessed. Indeed, global inhibition of PI3Ks (wortmannin, 1 $\mu$M), inhibition of mTOR

(temsirolimus, 3 $\mu$M), dual inhibition of PI3K and mTOR (dactolisib, 3 $\mu$M), and inhibition of MEK-1 (PD184352, 1 $\mu$M; cobimetinib, 0.5 $\mu$M) and ERK (SCH772984, 1 $\mu$M) led to an elevated 5-LO protein expression (1.2–2.8-fold, depending on the inhibitor) in HT-29 cells (Fig 2A). *ALOX5* mRNA levels were even more up-regulated (4.8–13-fold depending on the inhibitor) (Fig S5A). Inhibition of the EGF receptor (erlotinib, 5 $\mu$M) or the Ras farnesyltransferase (LB42708, 1 $\mu$M) induced a small increase in 5-LO expression as well, but this effect was not significant. In contrast, 5-LO protein expression in HCT-116 cells was only elevated upon treatment with inhibitors of the MEK-1/ERK cascade. Of note, the dual PI3K/mTOR inhibitor dactolisib led to a 2.6-fold induction of *ALOX5* mRNA, which was not significant

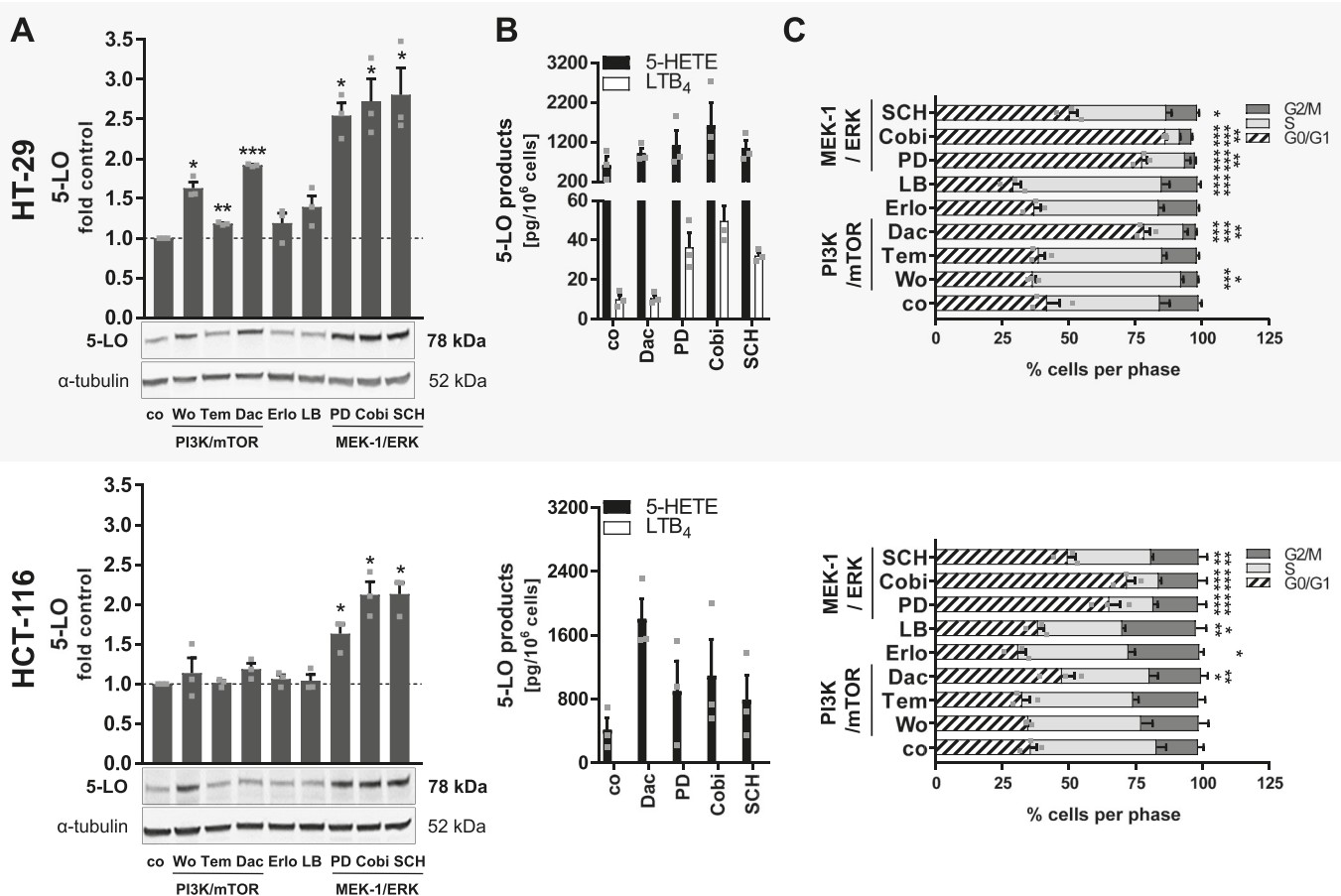

**Figure 2. 5-LO expression is up-regulated upon inhibition of enzymes involved in PI3K/mTOR and MEK-1/ERK signaling in HT-29 and HCT-116 cells.**
**(A)** 5-LO protein expression after 24 h of inhibitor treatment in HT-29 and HCT-116 cells grown as monolayers. Densitometric data were normalized to the loading control α-tubulin followed by normalization to the DMSO vehicle control (co). **(B)** 5-HETE and $LTB_4$ release from cells after 24 h of inhibitor treatment in HT-29 and HCT-116 cells grown as monolayers. Cells were stimulated with 2.5 $\mu M$ $Ca^{2+}$ ionophore (A23187) in the presence of 1 mM $CaCl_2$ (10 min, 37°C) and received exogenous ARA (20 $\mu M$). **(C)** Cell cycle distribution of HT-29 and HCT-116 cells after 24-h treatment with inhibitors. Cells were analyzed via flow cytometry. Data information: cells were cell cycle–synchronized by serum starvation 24 h before treatment. Results are depicted as the mean + SEM from three independent experiments. Squares indicate single data points. **(A, C)** Asterisks indicate significant changes versus DMSO vehicle control determined by a *t* test with Welch's correction (A) or two-way ANOVA coupled with Dunnett's post-test for multiple comparisons (C). *$P < 0.05$, **$P < 0.01$, and ***$P < 0.001$. Wortmannin (Wo), 1 $\mu M$; temsirolimus (Tem), 3 $\mu M$; dactolisib (Dac), 3 $\mu M$, erlotinib (Erlo), 5 $\mu M$; LB42708 (LB), 1 $\mu M$; PD184352 (PD), 1 $\mu M$; cobimetinib (Cobi), 0.5 $\mu M$; SCH772984 (SCH), 1 $\mu M$.

(Figs 2A and S5A). To confirm that this inhibitor-mediated elevation in 5-LO protein translated into higher 5-LO activity, cells were again treated with the inhibitors for 24 h. After this, cells were harvested and stimulated with $Ca^{2+}$ ionophore A23187 (2.5 $\mu M$) in the presence of exogenous ARA (20 $\mu M$) for 10 min. Depending on the inhibitor, formation of 5-HETE and $LTB_4$ was increased in HT-29 cells up to 2.6- and fivefold, respectively (Fig 2B). In HCT-116 cells, 5-HETE formation was up-regulated to a comparable extent, whereas $LTB_4$ was not released.

Having in mind the prominent growth arrest of HT-29 and HCT-116 monolayers after 7 d and the associated up-regulation of 5-LO, cell cycle distribution upon inhibitor treatment was measured (Fig 2C). Again, the extent to which 5-LO expression was induced roughly correlated with the level of G0/G1 arrest triggered by the inhibitors in both cell lines. Treatment with dactolisib (3 $\mu M$), PD184352 (1 $\mu M$), and cobimetinib (0.5 $\mu M$) induced a substantial shift toward the G0/G1 phase in HT-29 cells. Interestingly, this shift was less pronounced

for the ERK inhibitor SCH772984 (1 $\mu M$) (Fig 2C). In HCT-116 cells, inhibitors of the MEK-1/ERK axis potently induced a G0/G1 arrest, whereas inhibition of the PI3K/mTOR signaling did not affect the cell cycle (Fig 2C). Inhibitors that previously had only weak or no impact on 5-LO expression did not induce cell cycle arrest as well.

### p53 is not essential for 5-LO up-regulation upon spheroid growth and PI3K/mTOR and MEK-1/ERK inhibition

5-LO is known to be a direct p53 target gene, which is up-regulated via this pathway in HCT-116 cells during genotoxic stress (Gilbert et al, 2015). Because PI3K/mTOR and MEK-1/ERK signaling attenuated 5-LO protein expression, we were interested in whether this relates to p53 activity. Because HT-29 cells carry a mutated p53 (R273H) that lost its DNA binding capability, a direct influence of p53 on 5-LO expression in these cells was considered unlikely. To investigate whether the up-regulation of 5-LO in HCT-116 cells

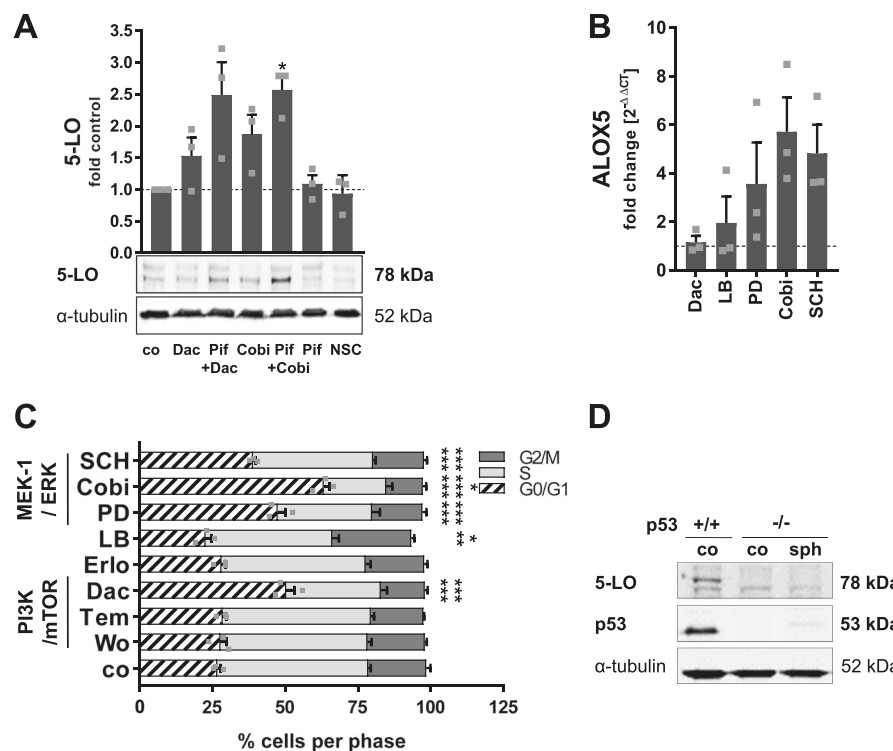

**Figure 3. Role of p53 in 5-LO up-regulation in MCTS and PI3K/mTOR and MEK/ERK inhibitor–treated monolayers of HCT-116 cells.**
**(A)** 5-LO expression in monolayers of HCT-116 WT cells treated with 3 μM dactolisib or 0.5 μM cobimetinib alone or in combination with the p53 inhibitor pifithrin-α (30 μM) for 24 h. Pifithrin-α and the MDM-2 inhibitor NSC68811 (5 μM) served as positive controls. One representative Western blot of three is shown. Densitometric data were normalized to the loading control α-tubulin followed by normalization to the DMSO vehicle control (co). **(B)** *ALOX5* mRNA expression in HCT-116 p53 −/− cells treated with various inhibitors for 24 h. Vehicle controls (co) received DMSO instead. Gene expression was normalized to *ACTB* and the respective vehicle control ($2^{-\Delta\Delta CT}$ method) after qRT-PCR analysis. **(C)** Cell cycle distribution of HCT-116 p53 −/− cells after 24-h treatment with inhibitors. Cells were analyzed via flow cytometry. **(D)** Protein expression of 5-LO and p53 in HCT-116 WT and p53 −/− cells grown as monolayers and MCTS for 7 d. Medium in the monolayer control (co) was changed daily, and cells were harvested after 48 h of culture to prevent cell stress. One representative Western blot of 4 is shown. Data information: cells were cell cycle–synchronized by serum starvation 24 h before treatment (A, B, C). Results are depicted as the mean + SEM from three independent experiments if not stated otherwise. **(A, B, C)** Squares indicate single data points. Asterisks indicate significant changes versus DMSO vehicle control determined by a *t* test with Welch's correction (A, B) or two-way ANOVA coupled with Dunnett's post-test for multiple comparisons (C). *$P < 0.05$, **$P < 0.01$, and ***$P < 0.001$. Wortmannin (Wo), 1 μM; temsirolimus (Tem), 3 μM; dactolisib (Dac), 3 μM, erlotinib (Erlo), 5 μM; LB42708 (LB), 1 μM; PD184352 (PD), 1 μM; cobimetinib (Cobi), 0.5 μM; SCH772984 (SCH), 1 μM.

induced by the treatment with the MEK-1/ERK inhibitors is p53-dependent, a number of assays were conducted. Basal 5-LO expression in HCT-116 monolayers was not influenced by the direct p53 inhibitor pifithrin-α (30 μM) (Fig 3A). Furthermore, interference with p53 degradation by inhibition of the E3 ubiquitin ligase MDM2 (NSC68811, 5 μM) had no impact on the levels of 5-LO in HCT-116 cells. When the cells were treated with the PI3K/mTOR inhibitor dactolisib (3 μM) or the MEK-1 inhibitor cobimetinib (0.5 μM) in the presence of pifithrin-α, 5-LO expression was further increased, suggesting a synergistic effect of p53 inhibition. Next, HCT-116 p53 −/− knockout cells, which show a reduced basal *ALOX5* expression on mRNA level and an absence of 5-LO protein in conventional monolayer culture, were treated with the PI3K/mTOR and MEK/ERK inhibitors for 24 h. This led to an increase in *ALOX5* mRNA expression comparable to the WT HCT-116 cells after treatment with the MEK-1 and ERK inhibitors PD184352 (1 μM), cobimetinib (0.5 μM), and SCH772984 (1 μM) (Fig 3B). Of note, 5-LO protein remained undetectable upon treatment with the inhibitors (Fig S5B). Furthermore, G0/G1 phase cell cycle arrest upon inhibitor treatment was also detectable in HCT-116 p53 −/− cells, although this effect was less pronounced compared with the cells carrying the WT p53 (Fig 3C). It is important to mention that the proportion of cells in the G0/G1 phase was lower from the start in HCT-116 p53 −/− KO cells compared with their WT counterparts. Finally, 5-LO expression in HCT-116 p53 −/− MCTS was investigated, showing that the elevated 5-LO expression upon three-dimensional growth is also affected by the KD (Fig 3D). These data show that p53 is not involved in the up-regulation of *ALOX5* mRNA upon inhibition of PI3K/mTOR and

MEK-1/ERK signaling but is important for mRNA translation to the 5-LO protein in HCT-116 cells.

## RICTOR and PI3-kinase catalytic subunits control 5-LO expression in HT-29 and HCT-116 cells

Because inhibition of PI3K/mTOR and MEK-1/ERK signaling potently elevated the expression of 5-LO in HT-29 and HCT-116 cells, we established a KD of central signaling components of both pathways in these cells to identify the central components involved. For this, both cell lines were transduced with lentiviral vectors coding for shRNAs directed against the different isoforms of the catalytic subunit of PI3K (p110α [*PI3KCA*], p110β [*PI3KCB*], p110δ [*PI3KCD*], and p110γ [*PI3KCG*]), Rictor (*RICTOR*), Raptor (*RPTOR*), mTOR (*MTOR*), and MEK-1 (*MAP2K1*). After confirming sufficient KD efficiencies, 5-LO expression was analyzed by Western blotting or qRT-PCR analysis. Against all expectations, KD of the mTOR kinase itself and the mTOR complex-1 adapter protein Raptor did only marginally influence 5-LO expression (Fig 4B and C). In contrast, KD of Rictor potently up-regulated 5-LO expression in both cell lines by 2.5 (HT-29)- and 2.9 (HCT-116)-fold, suggesting that the mTOR complex-2 (mTORC-2) fine-tunes 5-LO expression in HT-29 and HCT-116 cells instead (Fig 4A). Surprisingly, KD of different isoforms of the PI3K catalytic subunit showed that this enzyme has an opposing role in 5-LO expression in the cancer cells as reduction of p110α (*PI3KCA*) attenuated 5-LO levels in both cell lines (Fig 4E). Furthermore, KD of p110γ (*PI3KCG*) had a comparable effect in HT-29 cells (Fig S6). HCT-116 cells did not express this subunit on mRNA level. The PI3K p110β

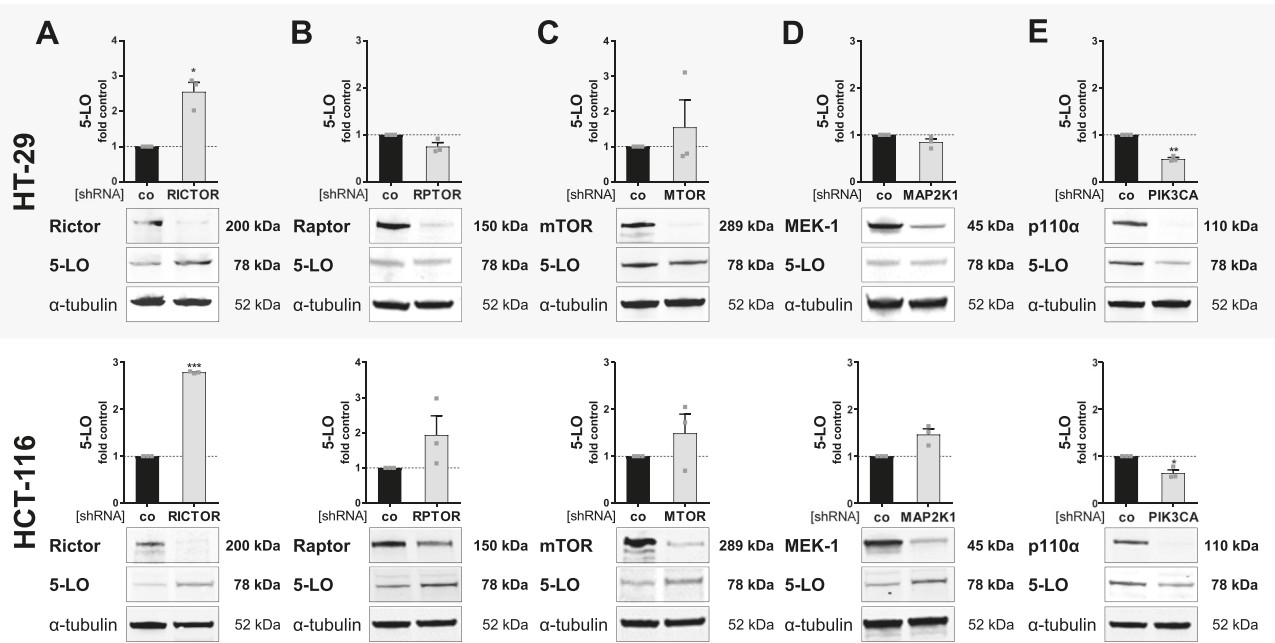

**Figure 4. RICTOR and PI3K p100α are involved in the control of 5-LO expression in HT-29 and HCT-116 cells.**
**(A, B, C, D, E)** 5-LO, and (A) Rictor, (B) Raptor, (C) mTOR, (D) MEK-1, and (E) PI3K p110α expression in stable HT-29 and HCT-116 knockdown cells versus control (shRNA co).
**(A, B, C, D, E)** Mean knockdown efficiencies; (A) HT-29: 49%, HCT-116: 36%; (B) HT-29: 79%, HCT-116: 37%; (C) HT-29: 87%, HCT-116: 73%; (D) HT-29: 58%, HCT-116: 56%; and (E) HT-29: 83%, HCT-116: 77%. Data information: densitometric data were normalized to the loading control α-tubulin followed by normalization to the non-mammalian shRNA-expressing control (co). Results are depicted as the mean + SEM from three independent experiments. Squares indicate single data points. One representative blot is shown. Asterisks indicate significant changes versus control determined by a *t* test with Welch's correction. *$P < 0.05$, **$P < 0.01$, and ***$P < 0.001$.

(*PI3KCB*) and p110δ (*PI3KCD*) subunits, and MEK-1 had no influence on 5-LO expression in both cell lines (Fig 4D and S6).

### Expression of 5-LO is subject to the cell cycle and locally limited to the quiescent viable zone of MCTS from HT-29 and HCT-116 cells

Because up-regulation of 5-LO coincided with treatments that induce a substantial G0/G1 cell cycle arrest and thus cell quiescence in HT-29 and HCT-116 cells, the influence of the cell cycle and cell cycle–related mediators was investigated. Initially, we analyzed the mRNA expression of several transcription factors that play a central role during cell cycle progression, among them forkhead box protein O(*FOXO*) 1 and 3, c-Myc (*MYC*), specificity protein 1 (*SP1*), and members of the DREAM (dimerization partner, RB-like, E2F, and multivulval class B) complex (*E2F1-3*, c-Myb [*MYB*], and b-Myb [*MYBL2*]) in cells treated with inhibitors of PI3K/mTOR and MEK-1/ERK signaling (Figs 5 and S6). In HT-29 cells, inhibition of MEK-1 and ERK significantly induced the expression of *FOXO3*, whereas the expression of *MYC* was significantly reduced. In contrast, the expression of *SP1* was up-regulated and *FOXO1* was not affected by the inhibitors (Fig 5A). In HCT-116 cells, *FOXO1*, *FOXO3*, and *SP1* were up-regulated by this treatment, whereas *MYC* was not affected (Fig 5A). The expression of members of the DREAM complex was also influenced by inhibition of MEK-1 and ERK. Here, *E2F1* and *MYBL2* were potently down-regulated in both cell lines. In HCT-116 cells also, the expression of *E2F2*, *E2F3*, and *MYB* was attenuated (Figs 5B and S7).

Treatment with dactolisib, the inhibitor of PI3K/mTOR signaling, led to an induction of *FOXO1*, *FOXO3*, *MYC*, and *SP1* in both cell lines, whereas the DREAM complex members were differently regulated.

In HT-29 cells, *E2F1*, *E2F2*, *MYB*, and *MYBL2* levels were attenuated (Fig 5A), whereas in HCT-116 cells, only *E2F2* and *MYBL2* were slightly down-regulated (Fig 5B). These data strongly suggest that 5-LO expression is under control of the cell cycle.

To prove this hypothesis, the cells were treated with the checkpoint inhibitors palbociclib (CDK4/6 inhibitor) and Ro-3306 (CDK1 inhibitor). Treatment with 1 μM palbociclib induced G0/G1 cell cycle arrest in HT-29 cells, whereas Ro-3306 (10 μM) triggered a G2/M arrest in both cell lines, as expected (Fig 6A). Because HCT-116 cells were less sensitive to CDK4/6 inhibition, a higher concentration of palbociclib was used (10 μM). Still, this only shifted the cells toward the G0/G1 phase, but failed to induce complete cell cycle arrest (Fig 6A). Then, 5-LO expression in the arrested cells was investigated. In HCT-116 cells, cell cycle arrest in G0/G1 and G2/M induced 5-LO expression, whereas in HT-29 cells, only G0/G1 arrest positively influenced the 5-LO level (Fig 6B).

Because 5-LO expression was positively correlated with G0/G1 arrest in HT-29 and HCT-116 cells, we postulated that the enzyme is induced upon cell quiescence/G0 arrest in the tumor spheroids. Therefore, the expression of cell cycle–associated transcription factors regulated upon inhibition of PI3K/mTOR and MEK-1/ERK in HT-29 and HCT-116 cells was investigated in MCTS and densely grown monolayers of both cell lines. MCTS formation and monolayer overgrowth significantly induced 5-LO expression on mRNA and protein levels for both cell lines as already described (Fig 1C and D). Along with 5-LO, *SP1* mRNA expression was induced in these incubations (Fig 6C and D) fitting the induction in *SP1* mRNA because of inhibition of PI3K/mTOR and MEK-1/ERK (Fig 5A). *E2F1* mRNA expression was only significantly down-regulated in the overgrown monolayers of both cell lines. Only in HT-29 MCTS grown for 7 d, a low reduction in *E2F1* expression was

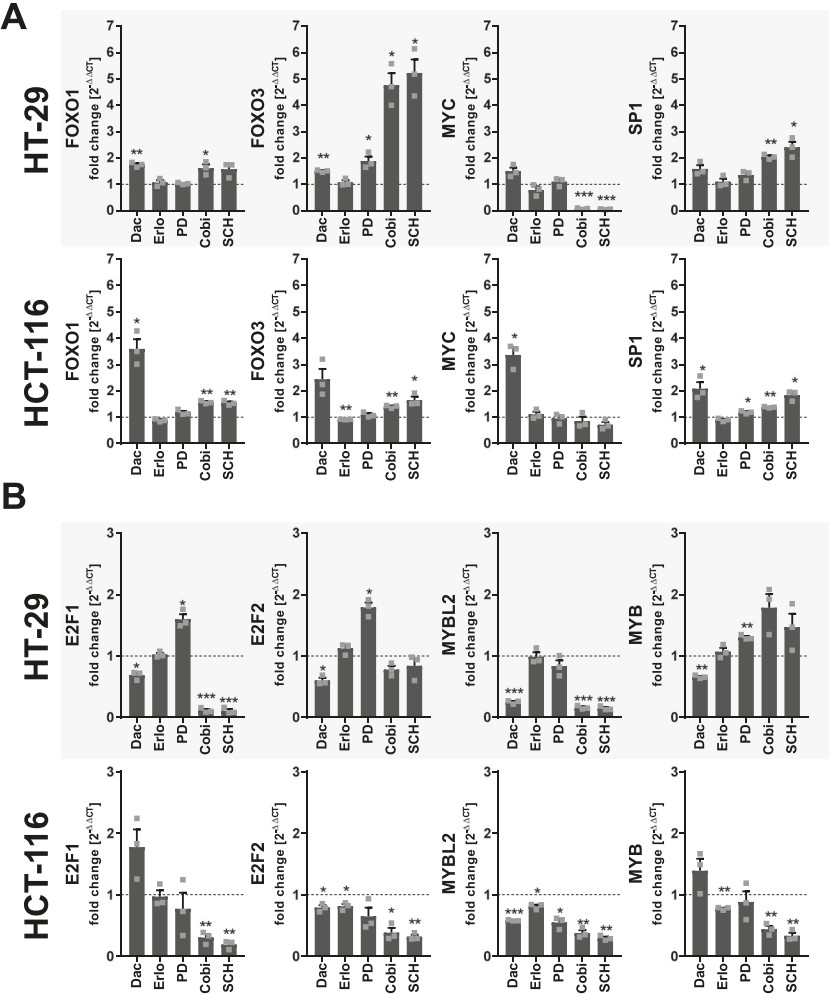

**Figure 5. Up-regulation of 5-LO expression after treatment with inhibitors of the PI3K/mTOR and MEK/ERK axis is accompanied by regulation of central transcription factors involved in cell cycle control in HT-29 and HCT-116 cells.** **(A, B)** Relative mRNA expression of (A) *FOXO1*, *FOXO3*, *MYC*, and *SP1* and (B) *E2F1*, *E2F2*, *MYBL2*, and *MYB* in HT-29 and HCT-116 cells treated with inhibitors for 24 h. Data information: cells were cell cycle–synchronized by serum starvation 24 h before treatment. Gene expression was normalized to *ACTB* and the respective vehicle control ($2^{-\Delta\Delta CT}$ method) after qRT-PCR analysis. Results are depicted as the mean + SEM from three independent experiments. Squares indicate single data points. Asterisks indicate significant changes versus DMSO vehicle control determined by a *t* test with Welch's correction. *$P < 0.05$, **$P < 0.01$, and ***$P < 0.001$. Dactolisib (Dac), 3 $\mu M$; erlotinib (Erlo), 5 $\mu M$; PD184352 (PD), 1 $\mu M$; cobimetinib (Cobi), 0.5 $\mu M$; SCH772984 (SCH), 1 $\mu M$.

observed (Fig 6C). *MYBL2* mRNA expression was reduced in both cell lines in overgrown monolayers and MCTS (Fig 6C and D), again in line with the regulation by the inhibitors (Fig 5B). Also, in line with the previously described inhibitor results was the observed *MYB* induction in HT-29 and the *MYB* reduction in HCT-116 MCTS and overgrown monolayer cells (Fig 6C and D).

Next, to investigate the distribution of 5-LO expression in the spheroid tissues and relate this to cell proliferation and death, cryosections of the MCTS were prepared. To differentiate between apoptotic and proliferating cells, the cryosections were stained for the proliferation marker Ki-67 or the apoptosis marker caspase-3 (cleaved) together with 5-LO. DAPI staining was used to visualize the cell nuclei. Subsequently, the sections were analyzed by confocal microscopy (Fig 7). HT-29 spheroids displayed a dense coherent tissue mass that showed a distinct necrotic core characterized by a large mass of fragmented nuclei in the DAPI staining. In contrast, HCT-116 cells formed less coherent, sometimes even hollow, structures. Ki-67 staining was mainly concentrated in the outer rim of the spheroid in both cell lines, as expected. In contrast, tissue areas that showed high caspase-3 cleavage were primarily found in the spheroid core or regions furthest from the surface.

Interestingly, spheroid formation did not change the distribution of 5-LO within the cells as the enzyme was still mainly found in the cytosol of both cell lines. In HT-29 spheroids, cells with high 5-LO expression were mainly found outside of the necrotic core region in the encasing viable mantle of cells interspersed with highly pro-liferating cells. Of note, the Ki-67–positive proliferating cells did not co-stain for 5-LO, suggesting that higher enzyme expression is found in quiescent cells. Interestingly, the indistinguishable dying cell mass of the necrotic core region also showed a stronger 5-LO staining. In HCT-116 cells, strong staining for 5-LO concentrated on the viable outer rim of the spheroid structures as well, but in contrast to HT-29 spheroids, the proliferating cells also expressed the enzyme.

### 5-LO expression is regulated in a E2F1- and b-Myb–dependent manner in HT-29 and HCT-116 cells

The expression of the DREAM complex components b-Myb and its regulator E2F1 was significantly attenuated upon treatment with the dual PI3K/mTOR inhibitor dactolisib and with two MEK-1 inhibitors (PD184352 and cobimetinib) and the ERK inhibitor SCH772984 in

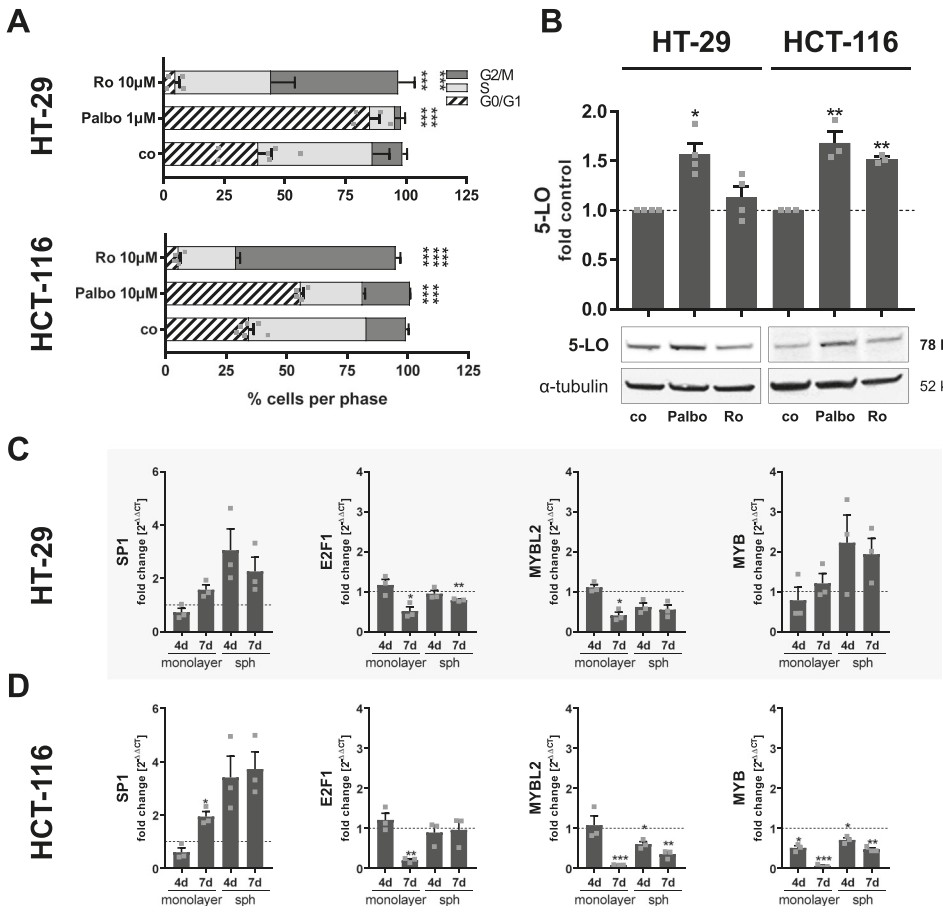

**Figure 6. Induction of cell cycle arrest induces 5-LO expression in HT-29 and HCT-116 monolayers.**

Accordingly, 5-LO up-regulation in MCTS is accompanied by regulation of central transcription factors involved in cell cycle control in both cell lines. **(A)** Cell cycle distribution of HT-29 and HCT-116 cells after 24-h treatment with inhibitors. Cells were analyzed via flow cytometry (control, n = 6; treatment, n = 4). **(B)** 5-LO protein expression after 24 h of inhibitor treatment in HT-29 and HCT-116 cells grown as monolayers. HT-29 cells were treated with 1 μM palbociclib or 10 μM Ro-3306. HCT-116 cells were treated with 10 μM palbociclib or 10 μM Ro-3306. Densitometric data were normalized to the loading control α-tubulin followed by normalization to the DMSO vehicle control (co). (HT-29, n = 4; HCT-116, n = 3). **(C, D)** mRNA expression of different cell cycle–associated transcription factors (*SP1*, *E2F1*, *MYBL2*, and *MYB*) in MCTS of HT-29 (C) and HCT-116 (D) cells grown for 4 and 7 d. Monolayer-grown cells (4 and 7 d) served as controls. Gene expression was normalized to *ACTB* and the respective monolayer control ($2^{-\Delta\Delta CT}$ method) after qRT-PCR analysis. Data information: cells were cell cycle–synchronized by serum starvation 24 h before the treatment (A, B). Results are depicted as the mean + SEM from three independent experiments if not stated otherwise. Squares indicate single data points. **(A, B, C, D)** Asterisks indicate significant changes versus control determined by two-way ANOVA coupled with Dunnett's post-test for multiple comparisons (A) or by a *t* test with Welch's correction (B, C, D). *$P < 0.05$, **$P < 0.01$, and ***$P < 0.001$. co, vehicle control; d, day; Palbo, palbociclib; Ro, Ro-3306; sph, MCTS.

HT-29 and HCT-116 monolayers. Furthermore, the expression of both factors was negatively correlated with 5-LO expression in the tumor spheroids. Because c-Myb, another member of the MYB family of transcription factors, is known to suppress 5-LO expression in undifferentiated HL-60 cells and in human macrophages during phagocytosis of tumor cells (Ponton et al, 1997; Ringleb et al, 2018), we further pursued the role of E2Fs and b-Myb in suppression of 5-LO in HCT-116 and HT-29 cells. For this, cells carrying a stable overexpression of b-Myb were generated. Unfortunately, both cell lines quickly silenced the overexpressed transgene. Therefore, cells carrying a doxycycline-inducible variant of b-Myb were generated instead. And indeed, we found a reduction in 5-LO protein expression 48 h after treatment with doxycycline (400 ng/ml HT-29, 200 ng/ml HCT-116) compared with the respective vector controls (Fig 8A). This reduction was more pronounced in HT-29 cells (40%) compared with HCT-116 cells (20%).

Two MYB binding elements have been characterized in the *ALOX5* gene so far, one upstream of the core promoter (binding sequence: ATAACGGTTTATT) (Samuelsson et al, 1991) and one within the coding sequence (predicted binding sequence: CAAAGTTG) (Ringleb et al, 2018). Therefore, cells carrying reporter constructs containing either the 5-LO core promoter (pN10LUC; –843 relative to the translation start [ATG]) or a larger promoter construct in which the putative MYB response element is situated (pN6LUC; –2,530 relative to the translation start [ATG]) were generated by stable transfection of

HT-29 and HCT-116 cells (Fig 8B). In addition, a reporter construct carrying a promoter derivative that lacked the MYB binding site (pN6ΔMYBLUC) was cloned. Firefly luciferase activity was normalized to constitutive EGFP expression derived from the same construct. The larger promoter pN6 showed a reduced relative firefly luciferase activity compared with the shorter pN10 construct in both cell lines confirming the presence of an element upstream of the core promoter that negatively regulates the ALOX5 gene in both cell lines (Fig 8C). Interestingly, mutation of the putative MYB binding site in pN6 led to a different outcome in HT-29 and HCT-116 cells. Deletion of the MYB binding site in HT-29 cells led to an activity loss of 83%, whereas in HCT-116 cells, pN6 promoter activity was increased by about 34% (Fig 8C). When HT-29 cells carrying the constructs for pN10LUC or pN6LUC were treated with the dual PI3K/mTOR inhibitor dactolisib or inhibitors of the MEK-1/ERK cascade (PD184352, cobimetinib, and SCH772984), a significant increase in relative firefly luciferase activity compared with vehicle (DMSO)-treated cells was detected that was lost in cells carrying the pN6 construct with deleted MYB binding site (pN6ΔMYB) (Fig 8D). In HCT-116 reporter cells, treatment with cobimetinib and SCH772984 led to a significant increase in relative firefly luciferase activity, whereas PD184352 only weakly induced the pN6 construct (Fig 8D). Deletion of the MYB response element did not influence the promoter activity in the inhibitor-treated cells. In line with the results on 5-LO expression upon dual PI3K/mTOR inhibition,

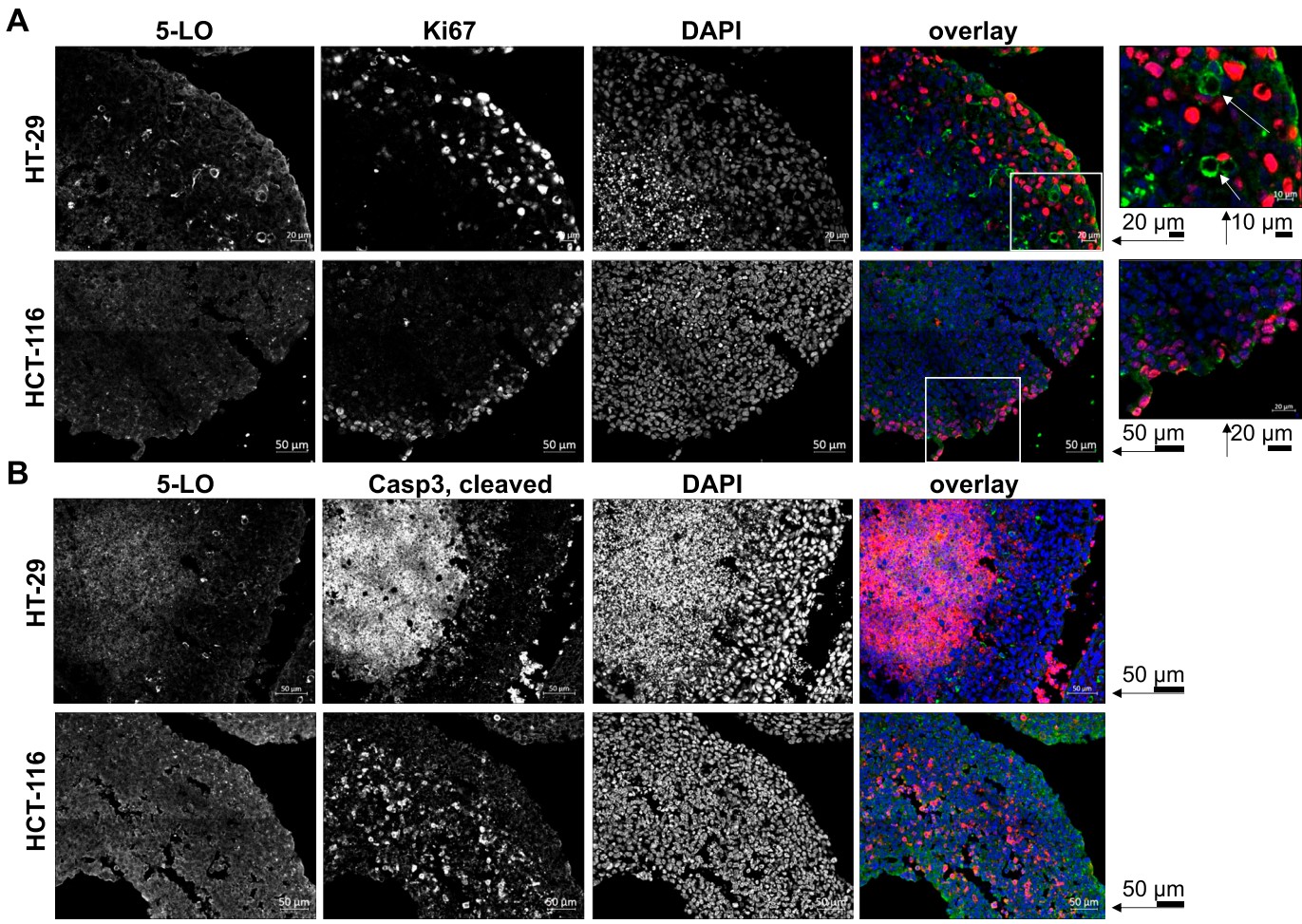

**Figure 7. Induction of 5-LO expression is locally limited to the quiescent viable zone of MCTS from HT-29 and HCT-116 cells.**
**(A, B)** Cryosections (14 μm) of HT-29 and HCT-116 MCTS grown for 7 d were co-stained with fluorophore-conjugated antibodies directed against (A) 5-LO (green) and the proliferation marker Ki-67 (red) or (B) 5-LO (green) and cleaved caspase-3 (red). Nuclear counterstaining was performed using DAPI (blue). Data information: the sections were analyzed by confocal microscopy using 3 × 3 (HT-29) or 4 × 4 (HCT-116) tile scans. Identical linear histogram adjustments were applied to each channel to adjust brightness and contrast. Displayed is a representative part of each digitally assembled scan. Single-channel fluorescence images are displayed in black and white for better contrast, whereas channel overlay is presented in color. Scale bars are provided within the figure for each row. One of three independent experiments is shown.

dactolisib failed to induce any *ALOX5* promoter construct tested in HCT-116 cells (Fig 8D). In addition to the *ALOX5* promoter reporter constructs, constructs containing the MYB binding site within the gene's coding sequence were generated. Unfortunately, all constructs containing the part of the coding sequence in which the second MYB binding element is situated showed only marginal firefly luciferase activity in both cell lines that was not significantly different to the control vector. Therefore, these constructs were not further evaluated.

We found that *E2F1* was down-regulated along with *MYBL2* in HT-29 and HCT-116 cells treated with inhibitors of PI3K/mTOR and MEK-1/ERK signaling. E2F1 is a transcription factor that is crucial for the regulation of cell cycle–dependent genes, among them *MYBL2* (Hanada et al, 2006; Sadasivam & DeCaprio, 2013). To further investigate the involvement of E2Fs in 5-LO expression, both cell lines were treated with the pan-E2F inhibitor HLM006474 (40 μM) for 24 h. Indeed, treatment of both cell lines led to a low, but in case of HCT-116 cells significant, up-regulation of 5-LO protein expression (Fig 8E).

Aberrant 5-LO expression is not only restricted to colorectal malignancies but also has frequently been found in other solid tumor entities and immortalized cell lines of these tumors. Therefore, we postulated that the regulation of 5-LO expression by PI3K/mTOR and MEK-1/ERK signaling is a mechanism that might apply to cancer cells from a solid origin in general. To test this hypothesis, we chose two cell lines that show substantial 5-LO expression (Capan-2 [adenocarcinoma, pancreas] and U-2 OS [osteosarcoma]) and two cell lines that are 5-LO–negative in our hands but enzyme expression has previously been reported by other groups (Caco-2 [adenocarcinoma, colon] and MCF-7 [adenocarcinoma, mammary gland]). All cell lines were cell cycle–synchronized for 24 h by serum starvation followed by treatment with the inhibitors (dactolisib, 3 μM [dual PI3K/mTOR]; PD184352, 1 μM [MEK-1]; cobimetinib, 0.5 μM [MEK-1]; and SCH772984, 1 μM [ERK]) in full-growth medium for further 24 h. Then, the mRNA expression of the *ALOX5* gene was assessed. In addition, we monitored the expression of *SP1*, *E2F1*, and *MYBL2* (b-Myb) because

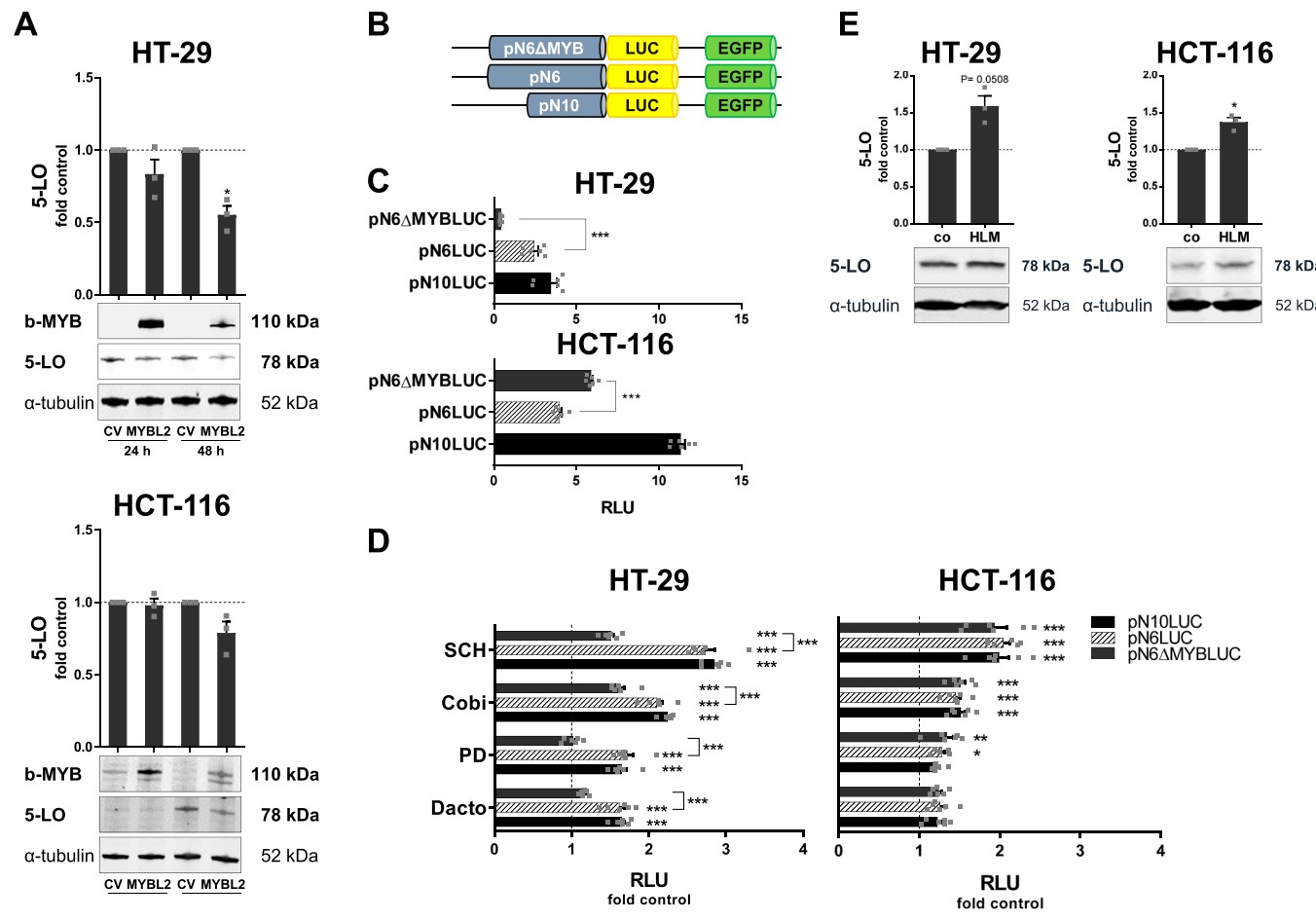

**Figure 8. 5-LO expression is regulated in a E2F1- and b-Myb–dependent manner in HT-29 and HCT-116 cells.**
**(A)** 5-LO protein expression after induction of b-MYB in cells carrying a doxycycline-inducible b-MYB (HT-29: 400 ng/ml doxycycline/24 h; HCT-116: 200 ng/ml doxycycline/48 h). Cells expressing the control vector construct served as controls (CV). Densitometric data were normalized to the loading control α-tubulin followed by normalization to the control vector. One representative Western blot of three is shown. **(B)** Schematic representation of the luciferase constructs carrying different parts of the 5-LO promoter (pN10LUC, −843 relative to the translation start [ATG], pN6LUC, −2,530 relative to translation start [ATG], pN6ΔLUC missing the 7-bp MYB binding site) used for the reporter gene experiments. **(C)** Basal activity of luciferase (LUC) reporter constructs in HT-29 and HCT-116 reporter cells. Results are given as RLU (normalized to EGFP) (n = 6). **(D)** Reporter gene assay in HT-29 and HCT-116 reporter cells after treatment with inhibitors of the PI3K/mTOR and MEK/ERK axis for 24 h. Results are depicted as fold vehicle (DMSO) control (n = 6). **(E)** 5-LO protein expression in HT-29 or HCT-116 monolayers treated with 40 µM of the E2F inhibitor HLM006474 (HLM) compared with vehicle-treated control cells (co). Densitometric data were normalized to the loading control α-tubulin followed by normalization to the DMSO vehicle control (co). One representative Western blot of three is shown. Data information: constructs were transfected in a stable manner in HT-29 and HCT-116 cells. For reporter gene assays and protein analysis, cells were subjected to cell synchronization using serum deprivation 24 h before treatment. Results are depicted as the mean + SEM. Squares indicate single data points. Asterisks indicate significant changes versus control determined by a *t* test with Welch's correction. *P < 0.05, **P < 0.01, and ***P < 0.001. Dactolisib (Dac), 3 µM; PD184352 (PD), 1 µM; cobimetinib (Cobi), 0.5 µM; SCH772984 (SCH), 1 µM. CV, control vector; EGFP, enhanced green fluorescent protein gene (*Aequorea victoria*); HLM, HLM006474; LUC, luciferase gene.

these transcription factors were prominently regulated along with *ALOX5* in HT-29 and HCT-116 cells.

Indeed, we found that both signaling cascades suppress the *ALOX5* gene in a number of other tumor cell lines. In Capan-2 cells, *ALOX5* mRNA expression was significantly up-regulated upon inhibition of the MEK-1/ERK signaling cascade (8.5–9.8-fold, depending on the inhibitor), whereas dual inhibition of PI3K/mTOR had only a weak effect (Fig 9A). Conversely, Caco-2 and MCF-7 cells up-regulated *ALOX5* mRNA upon inhibition of PI3K/mTOR (13.8- and 11.3-fold, respectively), whereas inhibitors of MEK-1/ERK signaling were not effective in these cells (Fig 9C and D). Interestingly, U-2 OS cells that already show high 5-LO expression were not responsive to any of the inhibitors (Fig 9B). In the three responsive cell lines

(Capan-2, Caco-2, and MCF-7), the regulation of *SP1*, *E2F1*, and *MYBL2* followed the pattern already seen in HT-29 and HCT-116 cells: up-regulation of *SP1* along with *ALOX5* and concomitant down-regulation of *E2F1* and *MYBL2* mRNA (Fig 9A–D). Interestingly, in U-2 OS cells the expression of *SP1* was not induced by any inhibitor although some of the treatments attenuated the *MYBL2* levels.

## Discussion

In contrast to 2D environments present in conventional cell culture, three-dimensional cell culture more closely resembles in vivo

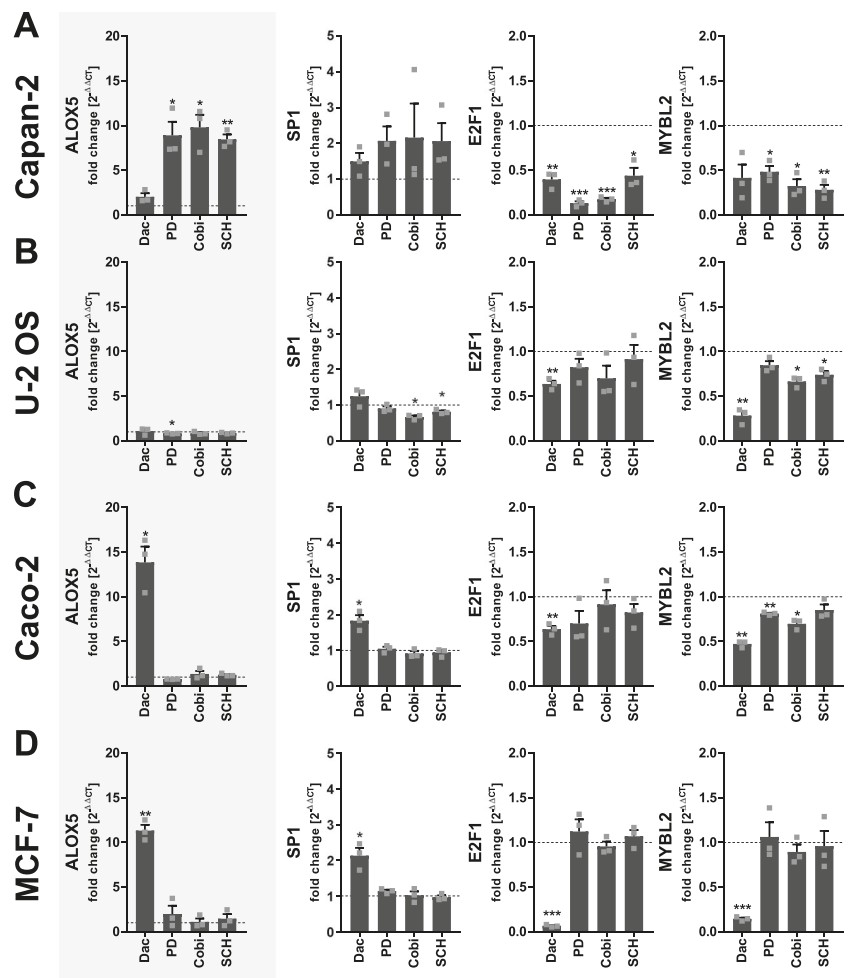

**Figure 9. Suppression of 5-LO by PI3K/mTOR and MEK/ERK-dependent signaling is a common trait among cancer cell lines derived from solid malignancies.**
**(A, B, C, D)** mRNA expression of *ALOX5*, *SP1*, *E2F1*, *MYBL2*, and Capan-2 (A), U-2 OS (B), Caco-2 (C), and MCF-7 (D) cells after 24-h treatment with inhibitors compared with vehicle controls (DMSO). Cells were subjected to cell synchronization using serum deprivation 24 h before treatment. Data information: gene expression was normalized to *ACTB* and the respective vehicle control ($2^{-\Delta\Delta CT}$ method) after qRT-PCR analysis. Results are depicted as the mean + SEM from three independent experiments. Squares indicate single data points. Asterisks indicate significant changes versus DMSO vehicle control by a *t* test with Welch's correction. *$P$ < 0.05, **$P$ < 0.01, and ***$P$ < 0.001. Dactolisib (Dac), 3 $\mu M$, PD184352 (PD), 1 $\mu M$; cobimetinib (Cobi), 0.5 $\mu M$; SCH772984 (SCH) 1 $\mu M$.

tissues. Here, cell–cell and cell–matrix interactions are present and gradients in cell proliferation and viability and in nutrients, catabolites, and oxygen are formed leading to fundamental changes in cell signaling (Lin et al, 2008; Ravi et al, 2015). In line with this, it has been recently shown that hyper-activated pro-proliferative signaling cascades such as PI3K/mTOR- and MEK-1/ERK-dependent pathways are potently down-regulated in tumor spheroids of colon cancer cells as a large proportion of cells in 3D structures are growth-arrested (Riedl et al, 2017). PI3K/mTOR- and MEK-1/ERK-dependent cascades are known to control a huge number of anabolic and catabolic processes. Thus, they are frequently hyper-activated in tumors of various origins (Sanchez-Vega et al, 2018). As a consequence, a substantial metabolic reprogramming of the malignant tissue takes place to meet the needs of the constantly proliferating cells (McCubrey et al, 2007; Papa et al, 2019).

In this work, we show that lipid mediator formation is affected by 3D growth of cancer cells in a PI3K/mTOR- and MEK-1/ERK-dependent manner. We found that the expression of 5-LO, $LTA_4H$, and $LTC_4S$, central enzymes in LT formation, is potently up-regulated when HT-29 and HCT-116 colon cancer cells are grown as MCTS. Our data are in line with a report showing that LT release from human glioma spheroids is four times higher compared with

monolayer-grown cells (Gáti et al, 1994). Using pharmacological inhibitors of members of the PI3K/mTORC and MEK-1/ERK cascade and RNAi technology, we found that 5-LO expression and activity were potently up-regulated in HT-29 and HCT-116 cells. These data suggest that both pathways attenuate 5-LO expression in the proliferating colon cancer cells. And indeed, we found that these pathways were also attenuated upon 3D growth of HT-29 and HCT-116 cells along with the up-regulation of 5-LO. Fig 10 comprehensively summarizes the signaling pathways that control 5-LO expression in solid cancer cells identified in the present study.

In the past, PI3K/mTOR signaling has already been shown to influence the formation of LTs on a post-transcriptional level in proliferating monocytes where it attenuates $LTC_4S$ activity via p70S6 kinase–mediated phosphorylation of the enzyme (Esser et al, 2011; Ahmad et al, 2016). Furthermore, PI3K/mTORC signaling is known to be involved in the reprogramming of breast cancer cells toward an oxylipin-dependent phenotype. These cells increase their de novo lipogenesis and the exogenous fatty acid uptake as a consequence of a mutated, constitutively activated PI3K catalytic p100$\alpha$ subunit (*PI3KCA*). Consequently, ARA is enriched and continuously released intracellularly in these cells, because of mTORC-2/PKC$\zeta$-dependent

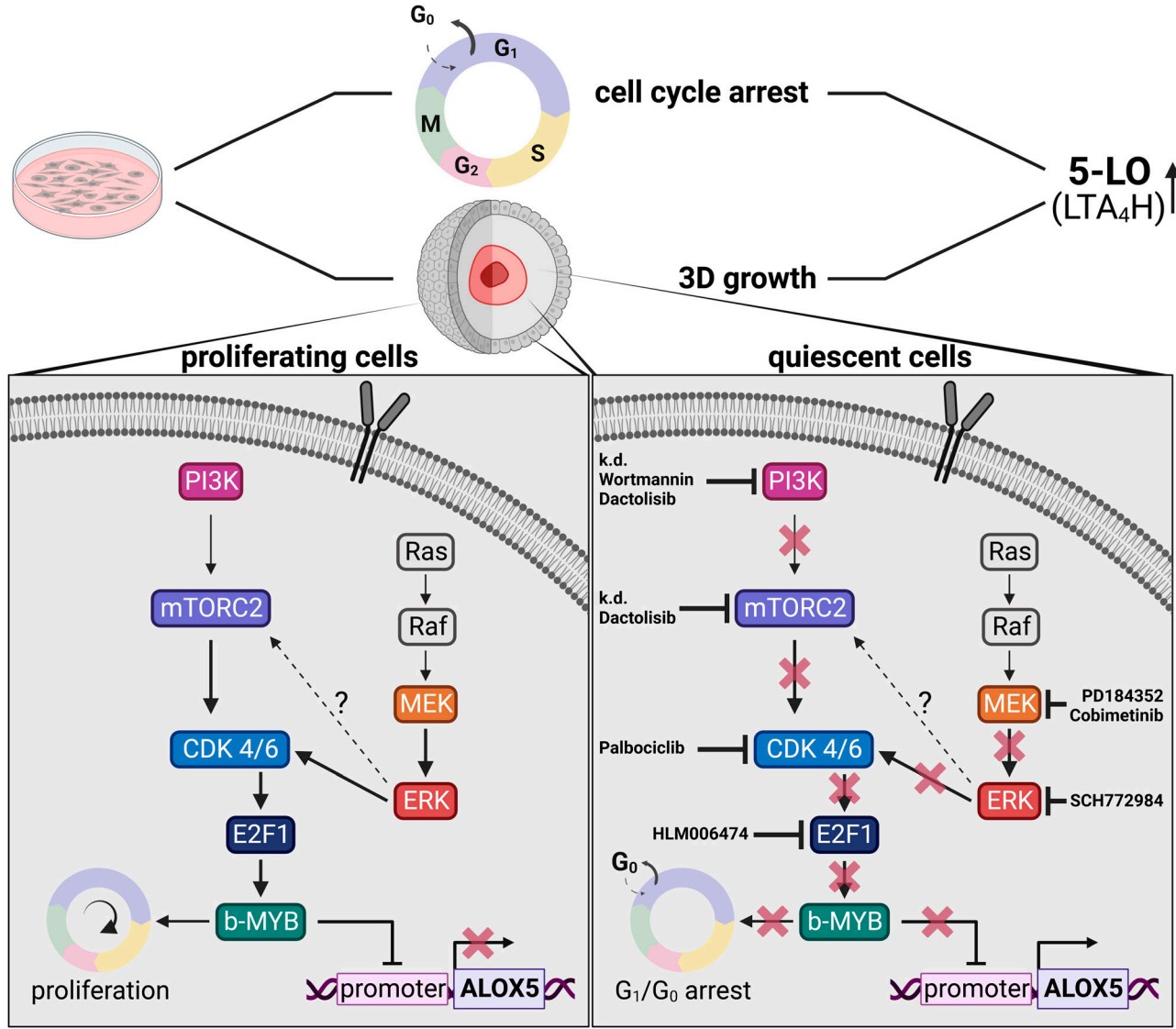

**Figure 10. Summary of the signaling pathways that control 5-LO expression in solid cancer cells identified in the present study.**

stabilization and activation of cPLA$_{2\alpha}$ (Koundouros et al, 2020). As a result, ARA-derived oxylipins with growth-stimulating properties such as 12- and 15-HETE are released acting in an autocrine manner on the cancer cells. Both HT-29 and HCT-116 cells also carry a mutated, constitutive active *PI3KCA*. This might explain why we found down-regulation of cPLA$_{2\alpha}$ in the MCTS of these cell lines along with attenuated PI3K/mTOR and MEK/ERK signaling and suggests that HT-29 and HCT-116 cells might also grow in an oxylipin-dependent manner. More importantly, our data show that in conditions where PI3K/mTORC-2 signaling is attenuated resulting in a down-regulation of cPLA$_{2\alpha}$ expression and cell cycle arrest, these oxylipin-dependent cancer cells elevate the expression and activity of LT cascade enzymes instead.

It is well established that 5-LO products stimulate the proliferation of cancer cells via activation of PI3K/AKT- and MEK/ERK-dependent cascades (Hii et al, 2001; Ding et al, 2003; Chang et al,

2019; Tang et al, 2021). It is therefore conceivable that under suboptimal growth conditions, oxylipin-dependent tumor cells down-regulate energy-expensive pro-proliferative pathways resulting in cell cycle arrest. We show here that at the same time, the central enzymes of the LT cascade, together with chemokines and growth factors such as IL-8 and VEGF, are up-regulated probably to attract and activate immune cells. Leukocytes such as platelets and granulocytes are capable of releasing huge amounts of ARA upon activation (Wesley Ely et al, 1995). The leukocyte-derived ARA might then compensate for the loss of cPLA$_{2\alpha}$ activity within the growth-arrested tumor cells to allow transcellular formation of 5-LO–derived oxylipins. This may represent a first step to restore cell proliferation in the tumor. Indeed, it has been reported years ago that growth-arrested glioma cells up-regulate 5-LO activity upon conditions that induce cell stress such as 3D growth and that these cells use 5-LO products to stimulate their own proliferation (Gáti et al,

1994). Furthermore, it has been recently shown that the expression of 5-LO in hypoxic areas of human ovarian tumors correlates with the number of tumor-associated macrophages and tumor progression (Wen et al, 2015).

Interestingly, proliferating solid cancer cells with constitutive 5-LO expression suppress LT formation by an as-yet-unknown mechanism (Weisser et al, 2022). Furthermore, we show here that a number of cancer cell lines that do not express 5-LO under conventional cell culture readily up-regulate the enzyme upon inhibition of PI3K/mTOR and/or MEK-1/ERK signaling. These data show that 5-LO activity is suppressed on several levels during cell proliferation. Interestingly, U-2 OS cells that already show a high basal 5-LO expression did not respond to the treatment with PI3K/mTOR and MEK-1/ERK inhibitors. These cells carry a *RICTOR* mutation (Tate et al, 2019), which explains their poor responsiveness and the particularly high basal 5-LO expression in these cells.

The readily scalable expression and activity of the LT pathway are remarkable and suggest that the presence of 5-LO is of advantage for tumor cells under cell stress conditions. Yet, why do tumor cells restrict 5-LO expression and activity during proliferation when the nutrient and oxygen demand is adequately addressed? This is probably due to the fact that reactive oxygen species generated during LT formation are a threat to cellular survival as they can covalently bind to nucleic acids or proteins (Hankin et al, 2003; Zhu et al, 2006) or induce cell death (Liu et al, 2015; Lee et al, 2022; Wu et al, 2022). Furthermore, the release of 5-LO products with chemoattractant properties such as $LTB_4$ and 5-oxo-ETE might interfere with the tumors' immune evasive tactic.

According to our data, cell cycle arrest and cellular quiescence play a fundamental role in the up-regulation of 5-LO expression in tumor cells. High 5-LO expression in tumor spheroids was inversely correlated with the presence of the proliferation marker Ki-67. Furthermore, 5-LO expression was up-regulated upon inhibition of central cell cycle–promoting factors such as CDK4/6 (palbociclib) and E2F1, whereas overexpression of b-Myb attenuated 5-LO levels in the colon cancer cells. In line with our data, it has been shown that cell cycle arrest induced by oncogenic RasV12 induces 5-LO in a number of cell lines. This was accompanied by an activation of p53 by 5-LO–derived ROS resulting in the activation of p53 and thus p21 (Catalano et al, 2005). Another correlation between cell proliferation and 5-LO expression has been reported for B cells (Werz et al, 2005). It was shown that proliferating cells express lower levels of 5-LO and this was attributed to a caspase-6–dependent cleavage of the enzyme during the cell cycle. Interestingly, the caspase-6 inhibitors used in this study also inhibit the caspase-3 and caspase-7 known to positively influence the cell cycle (Hashimoto et al, 2011). In light of our present results, it can be assumed that in addition to caspase cleavage, the induction of cell cycle arrest might also contribute to the control of 5-LO expression in B cells.

To our knowledge, this is the first time that negative regulation of 5-LO expression by b-Myb has been reported. In contrast, several studies could show that c-Myb, another member of the MYB family of transcription factors, can negatively influence 5-LO expression in myeloid cells. Depending on the study, the c-Myb effects were attributed to MYB consensus elements situated either in the *ALOX5* promoter region or within the coding sequence of the gene (Habenicht et al, 1989; Ponton et al, 1997; Ringleb et al, 2018). As

b-Myb and c-Myb share their DNA consensus element (Oh & Reddy, 1999), it is not surprising that the MYB binding elements responsible for the b-Myb–mediated suppression of 5-LO in the tumor cells are identical to the elements involved in c-Myb–mediated suppression of the enzyme in undifferentiated monocytic cells. Although our data comprehensively show that b-Myb is involved in the transcriptional repression of 5-LO during cancer cell proliferation, our data also suggest that additional factors control the expression of the enzyme.

5-LO expression is known to be regulated by epigenetic mechanisms involving histone deacetylases and polycomb repressive complex 2 members in myeloid cells (Klan et al, 2003; Wang et al, 2017). Therefore, transcription-repressing complexes controlling DNA accessibility might play an important role in the control of 5-LO expression in solid cancers. In addition to epigenetic regulation, a recent publication comprehensively showed that 5-LO is a direct p53 target gene in cancer, which is up-regulated upon treatment with cytostatic drugs (Gilbert et al, 2015). In line with this, another study reported the up-regulation of 5-LO along with p53 upon genotoxic stress but came to the conclusion that this is a p53-independent effect, as cells that carried a p53 frameshift mutation also up-regulated 5-LO upon these stimuli (Catalano et al, 2004). Our data can explain the discrepancies found in both studies because genotoxic stress is an inducer of cell cycle arrest.

Interestingly, in our hands elevation of *ALOX5* transcription was triggered in a p53-independent manner in cell cycle–arrested HCT-116 cells because inhibition of MEK/ERK signaling in p53−/−HCT-116 cells still induced the transcription of *ALOX5* mRNA. In contrast, 5-LO protein was not detectable in these cells, suggesting that p53 is involved in the translation of 5-LO upon inhibition of pro-proliferative pathways in HCT-116 cells. A number of p53 targets are indeed known to be influenced on a transcriptional level. Here, p53 regulates factors that control translation initiation of these targets (Marcel et al, 2018). HT-29 cells express a p53 mutant (R273H) that lost its DNA binding capacity. It is therefore unlikely that p53 is involved in the transcriptional control of the *ALOX5* gene in these cells.

Taken together, our data underline the importance of three-dimensional cell culture for investigations on the role of oxylipins in carcinogenesis. We show that a number of cancer cells derived from solid malignancies tightly regulate 5-LO expression, suppressing the enzyme during proliferation with the help of well-established pro-proliferative signaling pathways while readily up-regulating it upon cell stress conditions triggering cytostasis. This shows that although 5-LO expression is a potential threat to cancer cells, its expression might also be an opportunity for cell survival under stressful conditions because of the pro-proliferative properties of 5-LO products, the non-canonical functions of the enzyme that can influence key cancer cell functions, and a potential influence of these factors on the tumor stroma. Our data also suggest that cytostatic therapies might induce LT formation in growth-arrested cancer cells, thereby rescuing cell proliferation in the surviving tumor tissue.

Based on this, it seems obvious to inhibit LT formation along with cytostatic therapy in the treatment of solid cancers. Yet, inhibitors of LT formation and signaling lack clinical efficacy in cancer so far or even worsen the outcome of the patient (Edelman et al, 2008; Saif

et al, 2009; Bishayee & Khuda-Bukhsh, 2013; Jänne et al, 2014). This might be explained by the fact that in addition to its products, 5-LO can also influence tumor cell function in a non-canonical way. Furthermore, pharmacological inhibition of LT formation in the tumor is accompanied by a concurrent inhibition of LTs in leukocytes, which might interfere with the patients' anti-tumor response.

An answer to this problem might be the development of inhibitors that interfere with non-canonical functions of 5-LO but spare LT formation in leukocytes and thus the anti-tumor response of the immune system. Furthermore, further research should concentrate on fatty acid uptake, allocation, and LT release in oxylipin-dependent cancer cells and compare this with leukocytes to see whether differences exist that might be targeted to prevent the formation of oxylipins and their release from cancer cells. In addition, studies on the details of the cell cycle–dependent 5-LO regulation in cancer cells might help to develop strategies that can interfere with transcription/translation of the enzyme. And finally, studies further investigating the interplay between 5-LO–expressing tumors and cells of the tumor stroma might lead to the identification of possible targets.

# Materials and Methods

## Reagents and chemicals

Dulbecco's PBS, DMEM, McCoy's 5A (Modified) Medium, Opti-MEM I Reduced Serum Medium, sodium pyruvate (100 mM), penicillin–streptomycin (PenStrep, 10,000 U/ml penicillin, 10,000 $\mu$g/ml streptomycin), puromycin (10 mg/ml), Lipofectamine LTX Reagent with PLUS Reagent, PageRuler Prestained Protein Ladder, Spectra Multicolor High Range Protein Ladder, propidium iodide (PI), trypsin–EDTA solution (TE), *Power* SYBR Green PCR Master Mix, DNase I, RNase A (10 mg/ml), High-Capacity RNA-to-cDNA Kit, TRIzol, and the Pierce BCA Protein Assay Kit were purchased from Thermo Fisher Scientific or their associated companies Invitrogen or Applied Biosystems. Fetal calf serum (FCS) was obtained from Capricorn Scientific GmbH. PhosSTOP, cOmplete Mini (from Roche), MISSION shRNA pLKO.1-vectors, MEM non-essential amino acid solution (100×), polybrene, polyethylene glycol (PEG8000), polyethylenimine (PEI), coenzyme A (lithium salt), sodium hydrosulfite ($Na_2S_2O_4$), and adenosine 5'-monophosphate (disodium salt; AMP) were obtained from Sigma-Aldrich. Tris(hydroxymethyl)aminomethane (Tris), Tween-20, Triton X-100, Hepes, DTT, magnesium sulfate ($MgSO_2$), and sodium dodecyl sulfate were bought from PanReac AppliChem ITW Reagents. Sodium chloride (NaCl) and ethanol were purchased from Carl Roth. NEBuilder HiFi DNA Assembly Master Mix and all restriction enzymes used were purchased from New England Biolabs, Inc. Primers were obtained from Eurofins Scientific SE. Peroxide-free ARA, butylhydroxytoluol, cobimetinib, dactolisib, erlotinib, HLM006474, LB42708, PD184352, SCH772984, temsirolimus, pifithrin-α, NSC68811, and wortmannin were purchased from Cayman Chemical Company. Palbociclib and Ro-3306 were purchased by MedChemExpress. EDTA (Titriplex III) and α-D-glucose were purchased from Merck KGaA. Adenosine 5'-triphosphate (disodium salt; ATP) and calcium chloride were obtained from Carl Roth GmbH + Co. KG. Beetle luciferin (potassium salt; D-luciferin) was purchased from Promega Corporation.

## Cell culture

HT-29, HCT-116, MCF-7, Caco-2, and Capan-2 cells were obtained from DSMZ (Deutsche Sammlung von Mikroorganismen und Zellkulturen). U-2 OS cells were bought from ATCC (American Type Culture Collection). Lenti-X 293T cells were ordered from Takara Bio Inc. HCT-116 p53–/– cells were a kind gift from Bert Vogelstein (Johns Hopkins University). Cells were cultured in McCoy's 5A Medium (HT-29, Capan-2) or DMEM (HCT-116, HCT-116 p53–/–, MCF-7, Caco-2, and Lenti-X 293T) supplemented with 10% FCS, 1% sodium pyruvate, and 1% penicillin/streptomycin (complete growth medium, CGM). When cells were cultured in reduced growth medium (RGM), only 0.5% FCS was used. Caco-2 cells were also supplemented with 1% non-essential amino acids. All cells were regularly checked for mycoplasma contamination and cultured at 37°C and 5% $CO_2$ in a humidified atmosphere.

## Formation of MCTS

For spheroid formation, cells ($0.05 \times 10^6$ cells/200 $\mu$l CGM per well) were seeded into 96-well low adherence plates (CellCarrier Spheroid ULA Microplate, PerkinElmer) and allowed to grow for 4 and 7 d, depending on the assay (37°C and 5% $CO_2$ in a humidified atmosphere). For the respective monolayer controls, cells were seeded into six-well plates ($0.4 \times 10^6$ cells/2 ml CGM per well; mRNA analysis) or in 100-mm dishes ($3 \times 10^6$ cells/10 ml CGM; immunoblot analysis). Medium was changed after 24 h, and the cells were harvested after 48 h. 4-d and 7-d monolayer cells were seeded into 12-well plates ($0.05 \times 10^6$ cells/2 ml per well) with no medium change. Cellular morphology was monitored using a Zeiss Axio Vert.A1 microscope (Carl Zeiss AG) with an Axiocam 305 color. MCTS diameters were assessed using ZEN core 2.6 software (Carl Zeiss AG). For lipid mediator formation, $0.01 \times 10^6$ cells/well were seeded into 96-well low adherence plates (Corning 96-well Spheroid Microplates; Corning Incorporated) using the respective CGM. After 7 d, MCTS were harvested and washed.

## Inhibitor treatment

Cells were seeded into six-well plates ($0.4 \times 10^6$ cells/2 ml RGM; mRNA analysis) or 100-mm dishes ($3 \times 10^6$ cells/10 ml RGM; cell cycle, immunoblot, and lipid mediator analyses) to synchronize the cell cycle. After 22 h, medium was changed to CGM, 2 h before inhibitor treatment. Cells were treated with DMSO (control) or the respective inhibitors for 24 h. In addition, DMSO-treated controls of cells seeded in CGM were prepared.

## Generation of lentiviral KDs

For lentiviral particle production from MISSION shRNA pLKO.1-vectors (Table 1), Lenti-X 293T cells were seeded 24 h before transfection in 100-mm dishes ($4 \times 10^6$ cells in 10 ml CGM). Then, transfection mixes were prepared by mixing 10 $\mu$g MISSION shRNA pLKO.1-vectors (for further information on the plasmids, see

**Table 1.  shRNA knockdown target genes and respective sequences.**

| Target gene | Clone ID | TRCN | Binding region | shRNA sequence or target sequence (PI3KCB, PI3KCD, PI3KCG) |
|---|---|---|---|---|
| MAP2K1 | NM_002755.2-2032s1c1 | 0000199799 | 3′UTR | CCGGCCCATATCCAAGTACCAATGCCTCGAGG CATTGGTACTTGGATATGGGTTTTTTG |
| MTOR | NM_004958.2-4662s1c1 | 0000038678 | CDS | CCGGGCATGGAAGAATACACCTG TACTCGAGTACAGGTGTATTCTTCCATGCTTTTTG |
| PI3KCA | NM_006218.2-3471s1c1 | 0000196582 | 3′UTR | CCGGGCATTAGAATTTACAGCAA GACTCGAGTCTTGCTGTAAATTCTAATGCTTTTTTG |
| RICTOR | NM_152756.2-2620s1c1 | 0000074291 | CDS | CCGGCGTCGGAGTAACCAAAGATTAC TCGAGTAATCTTTGGTTACTCCGACGTTTTTG |
| RPTOR | NM_020761.1-4325s1c1 | 0000039770 | CDS | CCGGCGACTACTACATCTCCGTGTAC TCGAGTACACGGAGATGTAGTAGTCGTTTTTG |
| PI3KCB | NM_006219 | 0000010025 | CDS | CGACAAGACTGCCGAGAGATT |
| PI3KCD | NM_005026 | 0000033276 | CDS | GACCCAGAAGTGAACGACTTT |
| PI3KCG | NM_002649 | 0000199330 | CDS | CTCCAGATCTACTGCGGTAAA |

Table 1) with the plasmids for packaging and envelope (4.5 µg pCMV-VSV-G [plasmid #8454; Addgene]; 7.5 µg psPAX2 [plasmid #12260; Addgene]) in 500 µl Opti-MEM followed by dropwise addition of 500 µl PEI Mix (88 µg PEI in 500 µl Opti-MEM). The transfection mixes were incubated for 15 min at RT before addition to the cells. During transfection, the cells were kept in CGM containing 25 µM chloroquine for 17 h. For virus production, the transfection medium was replaced with 10 ml CGM and lentiviral particles released into the medium were harvested after 48 h. For this, the filtrated (0.45 µm, PVDF; Carl Roth GmbH + Co. KG) cell supernatants were mixed with 4× Lenti-X Concentrator (40% (W/V) PEG-8000, 1.2 M NaCl, pH 7.2) in a ratio of 3:1 followed by incubation for 30 min at 4°C. Then, the supernatants were centrifuged (1,500 rcf, 45 min, 4°C) and the resulting virus pellets were resuspended in 1 ml Opti-MEM.

For lentiviral transduction, HT-29 and HCT-116 cells were seeded into six-well plates ($0.3 \times 10^6$/well). After 24 h, the medium was replaced with 2 ml RGM supplemented with 8 µg/ml polybrene and 500 µl of the concentrated virus particles was added dropwise. Cells were then further incubated for 24 h. After this, the medium was changed to the respective CGM. After 48 h, puromycin (3 µg/ml) selection of the stably transduced cells was initiated. During the 12 d of selection, the medium was replaced every other day.

## Immunoblot analysis

For total cell lysates, the cells were suspended in lysis buffer (20 mM Tris–HCl, pH 7.4, 150 mM NaCl, 2 mM EDTA, 1% Triton X-100, and 0.5% NP-40) supplemented with protease and phosphatase inhibitors (PhosSTOP + cOmplete Mini), and kept on ice for 10 min followed by sonication and centrifugation (10,000 rcf, 10 min, 4°C) to remove insoluble debris. Then, total protein concentrations were assessed using the Pierce BCA Protein Assay Kit and measured with a multiplate reader (Infinite M200; Tecan Group Ltd.). Next, the cell lysates (40 µg/lane) were separated by sodium dodecyl sulfate–polyacrylamide gel electrophoresis followed by Western blot transfer to nitrocellulose membranes (0.2 µm; Bio-Rad) employing a wet tank method or the Trans-Blot Turbo Transfer System (both from Bio-Rad). PageRuler Plus Prestained Protein Ladder and Spectra Multicolor

High Range Protein Ladder were used as molecular weight markers. After blotting, the membranes were blocked with EveryBlot Blocking Buffer (Bio-Rad; 1 h, RT). After this, the membranes were incubated with primary antibodies directed against the indicated target genes (Table 2) followed by incubation with the respective fluorescence-conjugated secondary antibodies (IRDye; LI-COR Biosciences). α-Tubulin was used as a housekeeping control. Then, the protein antibody complexes were visualized with the Odyssey Infrared Imaging System (LI-COR Biosciences) and immunoreactive bands were quantified with Image Studio 5.2 software (LI-COR Biosciences; for information on the antibodies used, see Table 2).

## Immunohistochemistry

MCTS were harvested and washed with ice-cold PBS followed by fixation (4% PFA in PBS) for 30 min at RT. For cryopreservation, fixed MCTS were dehydrated with rising concentrations of sucrose (10–30%) for 30 min at 28°C, respectively. Dehydrated MCTS were snap-frozen in Tissue Freezing Medium (Leica Biosystems), and 14-µm

**Table 2.  Antibodies used for immunoblot analysis.**

| Primary antibodies | Cat. No. | Distributor |
|---|---|---|
| 5-LO | 66326-1-Ig | Proteintech |
| cPLA$_{2\alpha}$ | sc-438 | Santa Cruz Biotechnology |
| FLAP | ab53536 | Abcam |
| LTA$_4$H | sc-390567 | Santa Cruz Biotechnology |
| MEK-1 | #2352 | Cell Signaling Technology |
| mTOR | PA5-34663 | Thermo Fisher Scientific |
| p110α | MA5-14870 | Thermo Fisher Scientific |
| p53 | sc-126 | Santa Cruz Biotechnology |
| Raptor | sc-81537 | Santa Cruz Biotechnology |
| Rictor | sc-271081 | Santa Cruz Biotechnology |
| α-Tubulin | sc-5286 | Santa Cruz Biotechnology |
| α-Tubulin | ab176560 | Abcam |

cryosections were prepared using a CryoStar NX50 (Thermo Fisher Scientific). Tissue sections were then mounted to Superfrost Plus adhesion microscope slides (Epredia Netherlands B.V.) and dried for 1 h at RT. For staining, slides were washed in ice-cold PBS for 10 min. Tissue sections were blocked (1% BSA, 22.52 mg/ml glycine, and 0.2% Triton X-100 in PBS) for 1 h at RT. Then, tissues were incubated overnight at 4°C with the primary antibodies (Ki-67, ab15580, Abcam; cleaved caspase-3, #9664S, Cell Signaling Technology; and 5-LO, 66326-1-Ig; Proteintech) diluted in PBS containing 1% BSA and 0.2% Triton X-100. Then, samples were washed three times with ice-cold PBS and stained with fluorophore-coupled secondary antibodies (donkey anti-mouse IgG, Alexa Fluor 488, donkey anti-rabbit IgG, and Alexa Fluor 647; Thermo Fisher Scientific) for 1 h at RT. After washing three times with ice-cold PBS, samples were counterstained using 1 µg/ml DAPI (Thermo Fisher Scientific). Prepared samples were mounted (mounting medium: 0.1 g/ml Mowiol 4-88, 0.25 g/ml glycerol, and 25 mg/ml DABCO) followed by sealing using clear nail polish.

The slides were analyzed with a Zeiss 780 Axio Observer.Z1 laser scanning confocal microscope (Carl Zeiss AG), using a Zeiss Plan-Neofluar 40×/1.3 NA Oil lens. The same pinhole size was applied for all measurements. Pictures of HT-29 MCTS cryosections were obtained as 3 × 3 tile scans, whereas pictures of HCT-116 MCTS cryosections were obtained as 4 × 4 tile scans. Detailed acquisition parameters are provided in Table S1. Images were processed using ZEN Blue 2.6 software (Carl Zeiss AG).

## Cytometric bead array and ELISA

VEGF and IL-8 and RANTES (*CCL5*) concentrations in cell culture supernatants were measured using the BD Cytometric Bead Array. For this, BD Cytometric Bead Array Flex Sets (BD Biosciences) were used and the assay was performed according to the manufacturer's protocol using a BD FACSVerse flow cytometer. The resulting data were analyzed with FCAP Array 3.0 software.

TGF-β1 was measured with the TGF-β1 Human ELISA Kit (Invitrogen). For activation of latent TGF-β1, the cell culture supernatants were treated with 1 N HCl for 10 min at RT. After neutralization with 1 N NaOH, samples were measured according to the manufacturer's protocol. Finally, absorption was assessed at 450 and 570 nm (reference wavelength) using a microplate reader (Infinite 200; Tecan Group Ltd.).

## Cell cycle analysis

Inhibitor-treated cells were harvested after 24 h, and densely grown monolayer cells as described under "Formation of MCTS" were separated using a 20-µm cell strainer and spun down (340 rcf, 5 min, RT). For fixation, the resulting cell pellets were resuspended at a concentration of $4 \times 10^6$ cells/ml and 500 µl was added dropwise to 4.5 ml ice-cold ethanol under constant vortexing. Then, samples were stored overnight at −20°C. Before staining, fixed cells were washed twice with ice-cold PBS. Each pellet was then incubated with 500 µl staining buffer (PBS supplemented with 0.1% Triton X-100, 0.2 mg/ml RNase A, and 0.02 mg/ml propidium iodide) for 30 min at RT, light-protected. Cell cycle distribution was then measured with a FACS-Verse flow cytometer. For determination of single cells, sample sets

were gated by plotting PI-A/PI-W before applying the cell cycle analysis tool and applying the Watson (Pragmatic) model (FlowJo v10.6 software; Becton Dickinson BD Biosciences).

## mRNA isolation and quantitative RT–PCR

Total mRNA was extracted from cells using TRIzol reagent according to the manufacturer's protocol. After treatment with DNase I, mRNA was precipitated and purity and concentration were determined using UV/Vis spectroscopy (NanoDrop 2000; Thermo Fisher Scientific). In addition, mRNA integrity was investigated by agarose gel electrophoresis. Only samples showing sufficient mRNA integrity were reverse-transcribed (2 µg RNA) with the High-Capacity RNA-to-cDNA Kit according to the manufacturer's protocol. Relative quantification of mRNA content was performed by qRT-PCR analysis using 10 ng cDNA (RNA equivalent) with the *Power* SYBR Green PCR Master Mix (final volume 20 µl) on a StepOnePlus System (Applied Biosystems). Measurements were performed with three technical replicates per sample. β-Actin (*ACTB*) was used as a housekeeping gene for normalization. Primer sequences used in the experiments are shown in Table 3.

## Cloning of the pSBGP and pSBtet constructs and generation of stable reporter and b-Myb–overexpressing cells

Plasmids suitable for Sleeping Beauty transfection and protein expression in an inducible manner were prepared based on the pSBtet-GP backbone (plasmid #60495; Addgene), digested with NcoI and HindIII-HF. The *MYBL2* sequence was obtained by PCR amplification of a human cDNA clone (BioCat GmbH, Heidelberg, *MYBL2* sequence [Gene ID: 4605] in pCMV-SPORT6 vector), whereas a non-coding sequence for the empty vector control was obtained by PCR amplification of the MCS of a pSBbi-GH vector (plasmid #60514; Addgene). NEBuilder HiFi DNA Assembly Master Mix (NEB) was used to gain the final transfection plasmids pSBtet_MYBL2 and pSBtet_LV, respectively.

Reporter gene constructs carrying a firefly luciferase (FLuc) sequence were prepared using a modified pSBtet-GP backbone. For the 5-LO promoter constructs, the tetracycline-on system was removed by PCR amplification of the two remaining backbone parts. In addition, the FLuc sequence of the previously described plasmid pN10 was amplified by PCR. PCR products were assembled using the NEBuilder HiFi DNA Assembly Master Mix to gain the promoterless vector pSBGP-LUC. ALOX5 promoter sequences N10 (−843 relative to ATG) and N6 (−2,530 relative to ATG) were obtained by PCR amplification of the previously described plasmids pN10 and pN0 (Sorg et al, 2006), respectively. The pSBGP_LUC backbone was linearized by PCR, and the final plasmids (pSBGP_pN10LUC; pSBGP_pN6LUC) were obtained using the NEBuilder HiFi DNA Assembly Master Mix. To generate the pSBGP_pN6ΔMYBLUC plasmid, deletion of an MYB binding site within the pN6 promoter region was necessary. Deletion was achieved by restriction digestion of the pSBGP_pN6LUC vector using EcoRV-HF and PvuI-HF (NEB) and PCR amplification of both the upstream and downstream regions of the binding site. Assembly using NEBuilder HiFi DNA Assembly Master Mix yielded the final pSBGP_pN6ΔMYBLUC plasmid. All plasmid constructs were verified by DNA sequencing.

**Table 3. Gene-specific primer pairs.**

| Gene | Forward primer 5′-3′ | Reverse primer 5′-3′ | Nucleotide accession number | Product size [bp] |
|---|---|---|---|---|
| ALOX5 | CCC GGG AGA TGA GAA CCC TA | CCA GCA GCT TGA AAA TGG GG | NM_000698.5 | 200 |
| ACTB | AGA GCT ACG AGC TGC CTG AC | AGC ACT GTG TTG GCG TAC AG | NM_001101.5 | 184 |
| SP1 | AGT TCC AGA CCG TTG ATG GG | GTT TGC ACC TGG TAT GAT CTG T | NM_138473.3 | 101 |
| MYB | GCA GGT GCT ACC AAC ACA GA | CGA GGC GCT TTC TTC AGA TA | NM_001130173.2 | 175 |
| MYBL2 | CCA GCC ACT TCC CTA ACC G | CAG TGT CCA CTG CTT TGT GC | NM_002466.4 | 152 |
| E2F1 | GAG GAG ACC GTA GGT GGG AT | GGA CAA CAG CGG TTC TTG C | NM_005225.3 | 183 |
| E2F2 | GAG TCA GAG GAT GGG GTC CT | AAA CAT TCC CCT GCC TAC CC | NM_004091.4 | 156 |
| E2F3 | GGA GCT AGG AGA AAG CGG TC | TGA GGG AGA TTT TGG AGT TTT GG | NM_001949.5 | 115 |
| TP53 | CTG GAT TGG CAG CCA GAC T | TCC GGG GAC AGC ATC AAA TC | NM_000546.6 | 180 |
| FOXO3 | CGG ACA AAC GGC TCA CTC T | GGA CCC GCA TGA ATC GAC TAT | NM_001455.4 | 150 |
| FOXO1 | TCG TCA TAA TCT GTC CCT ACA CA | CGG CTT CGG CTC TTA GCA AA | NM_002015.4 | 168 |
| BAX | CCC GAG AGG TCT TTT TCC GAG | CCA GCC CAT GAT GGT TCT GAT | NM_001291430.2 | 155 |
| MYC | GTC AAG AGG CGA ACA CAC AAC | TTG ACG GAC AGG ATG TAT GC | NM_002467.6 | 162 |
| PI3KCB | TAT TTG GAC TTT GCG ACA AAG ACT | TCG AAC GTA CTG GTC TGG TAG | NM_006219.3 | 190 |
| PI3KCD | AAG GAG GAG AAT CAG AGC GTT | GAA GAG CGG CTC ATA CTG GG | NM_005026.5 | 138 |
| PI3KCG | GGC GAA ACG CCC ATC AAA AA | GAC TCC CGT GCA GTC ATC C | NM_002649.3 | 150 |
| LTC4S | GTC TAC CGA GCC CAG GTG AA | GCG TAG CCC TGG AAG TAG C | NM_145867.1 | 149 |

To generate stable cell lines, the Sleeping Beauty (SB) co-transfection system was used. For each construct, HT-29 (0.5–0.7 × 10⁶/3 ml) or HCT-116 (0.5–0.65 × 10⁶/3 ml) cells were seeded into six-well microplates. After 24 h, the medium was changed to 2 ml Opti-MEM per well. Then, stable transfection was performed for 24 h using the Lipofectamine LTX Reagent according to the manufacturer's protocol (LTX/DNA ratio: HT-29, 1:5; HCT-116, 1:3). All DNA mixes contained the SB100X transposase plasmid (kindly provided by Zoltan Ivics, Paul-Ehrlich-Institut) in a ratio of 1:20. After this, the medium was changed to the respective CGM. After another 24 h, puromycin (3 μg/ml) selection of the stably transfected cells was initiated. During the 9 d of selection, the medium was replaced every other day.

**Reporter gene assay**

0.03 × 10⁶ reporter cells per well were seeded into black 96-well μCLEAR plates (Greiner Bio-One) using 100 μl RGM. After 22 h, 100 μl CGM with 17.5% FCS was added to each well for 2 h. After that, the medium was changed to CGM containing the respective inhibitors or DMSO. After 24 h, firefly luciferase (FLuc) activity and EGFP expression were measured using a multiplate reader (Spark, Tecan Group Ltd.). For this, the cells were washed with PBS. Subsequently, the GFP signal was measured in 75 μl PBS/well (excitation wavelength: 485[±20]; emission wavelength: 520[±10]; optimal gain). Then, 75 μl assay buffer (74.9 mM Hepes, pH 7.8, 49.9 mM DTT, 4 mM MgSO₄, 895 μM AMP, 785 μM EDTA, 488 μM ATP, 469 μM D-luciferin potassium salt, 287 μM NaS₂O₄, 135 μM coenzyme A, 0,33% Tween-20, and 1% Triton X-100) was added per well. The culture plates were agitated at a low speed for 20 min at RT, and 140 μl of each well was transferred into a white LUMITRAC 96-well

plate (Greiner Bio-One). The FLuc activity was detected with an integration time of 1,000 ms. For analysis, the ratio of the FLuc to EGFP signals was determined and results were normalized to the respective DMSO control.

**Lipid mediator formation**

Preformed MCTS were harvested, washed, and resuspended in 500 μl PGC buffer (PBS, 1 mg/ml glucose, and 1 mM calcium chloride). In incubations with digested MCTS, spheroids were incubated in 200 μl Accutase (30 min, 37°C) to separate the cells. Consequently, cell numbers were assessed and the cells were resuspended in 500 μl in ice-cold PGC buffer after centrifugation (872 rcf, 5 min, 4°C). Cells grown as subconfluent monolayers for 4 d served as a control. For lipid mediator formation after inhibitor treatment, cells were harvested, washed, and resuspended in ice-cold PGC buffer at a concentration of 10⁷ cells/ml. Lipid mediator formation was initiated by stimulation with 2.5 μM Ca²⁺ ionophore (A23187) in the presence of 20 μM ARA. After 10 min, the reactions were stopped by the addition of 500 μl (MCTS) or 1,000 μl (cell suspensions) ice-cold methanol. In the MCTS assays, equal cell numbers were used for monolayer controls and the spheroids. In addition, the total protein amount per sample was analyzed for normalization.

Lipid mediators in the assay supernatants were analyzed by liquid chromatography–tandem mass spectrometry (LC-ESI-MS/MS). Therefore, samples were blinded. In brief, 200 μl sample was spiked with 20 μl MeOH, 20 μl methanolic IS working solution, 100 μl 0.15 M EDTA solution, 20 μl methanolic IS working solution, and 100 μl 0.15 M EDTA solution. These mixtures were extracted twice with 600 μl ethyl acetate. The organic phase was combined

after vortexing with subsequent centrifugation (3 min at 20,000 rcf) and evaporated at 45°C under a gentle stream of nitrogen. The extract was reconstituted in 50 $\mu$l MeOH/water (70:30, vol/vol) containing 0.0001% butylhydroxytoluol, and injected into the LC-MS system. Calibration standards and quality control samples were prepared by spiking 200 $\mu$l surrogate matrix with 20 $\mu$l of the methanolic standard working solution and processed as described for the samples.

The LC-MS system consisted of a triple quadrupole mass spectrometer QTRAP 6500+ (Sciex) equipped with a Turbo V ion spray source operated in a negative electrospray ionization mode and an Agilent 1,290 Infinity LC System (Agilent). The chromatographic separation was performed using an ACQUITY UPLC BEH C18 2.1 × 100 mm column for reversed-phase separation, and a Lux Amylose-1 column (250 × 4.6 mm, 3 $\mu$m, 1,000 Å, from Phenomenex) for chiral separation. Analytes were eluted with gradient elution using water (solvent A) and acetonitrile (solvent B) with 0.0025% formic acid, respectively. Data acquisition was done using Analyst software 1.7.1, and quantification was performed with MultiQuant software 3.0.3 (both from Sciex), employing the internal standard method (isotope dilution mass spectrometry). Calibration curves were calculated by linear regression with 1/x weighting.

## Statistical analysis

All data are presented as the mean + SEM. Statistical analysis was performed using GraphPad Prism 7. Data were subjected to an unpaired $t$ test with or without Welch's correction or two-way ANOVA coupled with Dunnett's post-test for multiple comparisons.

# Data Availability

This study includes no data deposited in external repositories. Data sets and biological materials will be provided upon request.

# Supplementary Information

# Acknowledgements

The authors thank Professor Dr. Robert Fürst for providing the flow cytometry instrument and Professor Dr. Dr. Achim Schmidtko for providing the cryostat. This work was supported by grants from the German Research Foundation (DFG; SFB1039 Z01 and GRK2336 TP4).

## Author Contributions

T Göbel: conceptualization, investigation, visualization, methodology, and writing—original draft, review, and editing.
B Goebel: investigation, methodology, and writing—original draft, review, and editing.
M Hyprath: investigation, methodology, and writing—review and editing.
I Lamminger: investigation.
H Weisser: validation and methodology.
C Angioni: validation, investigation, methodology, and writing—review and editing.
M Mathes: investigation and writing—review and editing.
D Thomas: validation, methodology, and writing—review and editing.
AS Kahnt: conceptualization, supervision, visualization, methodology, project administration, and writing—original draft, review, and editing.

## Conflict of Interest Statement

The authors declare that they have no conflict of interest.

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
