## [Reviewer comments · Life Science Alliance]

Life Science Alliance

Three-dimensional growth reveals fine-tuning of 5-lipoxygenase by proliferative pathways in cancer.

Astrid Kahnt, Tamara Göbel, Bjarne Goebel, Marius Hyprath, Ira Lamminger, Hannah Weisser, Carlo Angioni, Marius Mathes, and Dominique Thomas

DOI: <https://doi.org/10.26508/lsa.202201804>

Corresponding author(s): Astrid Kahnt, Goethe University Frankfurt

Review Timeline:

Submission Date:	2022-11-07
Editorial Decision:	2022-12-08
Revision Received:	2023-01-25
Editorial Decision:	2023-02-08
Revision Received:	2023-02-15
Accepted:	2023-02-16

Scientific Editor: Novella Guidi

Transaction Report:

December 8, 2022

Re: Life Science Alliance manuscript #LSA-2022-01804-T

Dr. Astrid S Kahnt
Goethe University
Institute of Pharmaceutical Chemistry
Max-von-Laue-Straße 9
Frankfurt 60438
Germany

Dear Dr. Kahnt,

Thank you for submitting your manuscript entitled "Three-dimensional growth reveals fine-tuning of 5-lipoxygenase by proliferative pathways in cancer." to Life Science Alliance. The manuscript was assessed by expert reviewers, whose comments are appended to this letter. We invite you to submit a revised manuscript addressing the Reviewer comments.

Thank you for this interesting contribution to Life Science Alliance. We are looking forward to receiving your revised manuscript.

Sincerely,

B. MANUSCRIPT ORGANIZATION AND FORMATTING:

Reviewer #1 (Comments to the Authors (Required)):

In the present manuscript, Globel et al. aimed to study the regulation of 5-LO in cancer cells. In particular, the authors investigated the expression of 5-LO and other enzymes involved in LT biosynthesis and lipid mediator formation in 5-LO-expressing tumor cell lines cultured as MCTS (multicellular colon tumor spheroids) and conventional cell culture conditions. Two 5-LO-positive colon carcinoma cell lines, HT-29 and HCT-116, were studied. They found that 5-LO expression is potently upregulated upon 3D growth in both cell lines. This upregulation was inversely correlated with cell proliferation and activation of PI3K/mTORC-2- and MEK-1/ERK-dependent pathways. E2F1 and its target gene MYBL2 were involved in the repression of 5-LO during cell proliferation. PI3K/mTORC-2- and MEK-1/ERK-dependent suppression of 5-LO also occurred in tumor cells from other origins.

This is an interesting and well-performed study. Many experiments were performed utilizing appropriate techniques. This study shows the relevance of using three-dimensional cell cultures to investigate the role of the eicosanoids and the regulation of the expression of genes encoding the enzymes involved in their biosynthesis. The findings of cell cycle-dependent 5-LO regulation in cancer cells are of interest. They performed many experiments to clarify the molecular mechanisms involved in it.

I have raised some minor points.

1. The introduction and, in particular, the discussion is too long. Thus, the manuscript can improve by shortening these parts.
2. It would be important to prepare a figure with a cartoon summarizing the pathways discovered in the control of 5-LO regulation.
3. Report a possible cancer treatment strategy utilizing the present manuscript's findings.
4. The authors show that the upregulation of 5-LO was inversely correlated with cell proliferation and activation of PI3K/mTORC-2- and MEK-1/ERK-dependent pathways. Was this response associated with the upregulation of the cyclooxygenase pathway? The authors can comment on this possibility in the discussion.
5. The authors can comment on the impact of 5-LO inhibitors and receptor antagonists on cancer development in light of their findings.

Referee Cross-Comments

Comments on referee # 2 concerns

There are several lines of evidence supporting the role of 5-LOX in cancer. It is relevant to perform studies focusing on this topic.

Comments on referee #3 concerns

The use of Spheroids is interesting because there is a need to develop alternative models to the use of experimental animal models. Thus, I found that the experimental approach followed by the authors is original. I agree that the manuscript is too lengthy.

Reviewer #2 (Comments to the Authors (Required)):

This article is of potentially great interest but first some basic questions need to be addressed. The findings in this report should be divided into several separate papers.

Major points: It is still very unclear whether cells that seem to contain a silent "5-LOX", really contain a full-length enzyme. The first point should be to isolate and sequence the enzyme from one of these cell lines. The authors have meritoriously found a way to increase the expression of the enzyme. Second major point. It is also very unclear which role 5-LOX plays in cancer. The authors must be very careful when they state that 5-LOX plays a role in cancer. Today we only know that certain B-cell lymphoma have high expression of 5-LOX; other single publications about 5-LOX and cancer have not yet been confirmed by other independent researchers. Furthermore, there seems to be a substantially auto-oxidative formation of 5-HETE in several experiments.

Reviewer #3 (Comments to the Authors (Required)):

In this article, Gobel et al. showed how lipooxygenase-derived lipid mediator biosynthesis is also affected by 3D growth. They

have studied the 5-LOX regulation and effects on several pathways in experiments using both 2D and 3D cultures of two colon cancer cell lines. A number of in vitro experiments were conducted, and the data is well presented. However, this reviewer does not see the novelty in the research. All the studies performed are in vitro (Spheroids). Moreover, the manuscript is too lengthy.

Reviewer 1:

All changes in the manuscript related to reviewer 1 are high-lighted in red.

In the present manuscript, Göbel et al. aimed to study the regulation of 5-LO in cancer cells. In particular, the authors investigated the expression of 5-LO and other enzymes involved in LT biosynthesis and lipid mediator formation in 5-LO-expressing tumor cell lines cultured as MCTS (multicellular colon tumor spheroids) and conventional cell culture conditions. Two 5-LO-positive colon carcinoma cell lines, HT-29 and HCT-116, were studied. They found that 5-LO expression is potently upregulated upon 3D growth in both cell lines. This upregulation was inversely correlated with cell proliferation and activation of PI3K/mTORC-2- and MEK-1/ERK-dependent pathways. E2F1 and its target gene MYBL2 were involved in the repression of 5-LO during cell proliferation. PI3K/mTORC-2- and MEK-1/ERK-dependent suppression of 5-LO also occurred in tumor cells from other origins.

This is an interesting and well-performed study. Many experiments were performed utilizing appropriate techniques. This study shows the relevance of using three-dimensional cell cultures to investigate the role of the eicosanoids and the regulation of the expression of genes encoding the

enzymes involved in their biosynthesis. The findings of cell cycle-dependent 5-LO regulation in cancer cells are of interest. They performed many experiments to clarify the molecular mechanisms involved in it.

I have raised some minor points.

1. The introduction and, in particular, the discussion is too long. Thus, the manuscript can improve by shortening these parts.

We have shortened the introductory part. In addition, the discussion section was substantially reduced.

2. It would be important to prepare a figure with a cartoon summarizing the pathways discovered in the control of 5-LO regulation.

We have now added an additional figure (Fig. 10) with a cartoon depicting the findings of the present manuscript.

3. Report a possible cancer treatment strategy utilizing the present manuscript's findings.

We respond here to the points 3 and 5 raised by reviewer No.1 in a combined way:

Based on the already existing literature in combination with the data presented here, it seems obvious to inhibit LT formation along with cytostatic therapy since cell cycle arrest and probably 5-LO expression is triggered in the cancer cells during treatment. Unfortunately, this is wishful thinking at the moment due to a number of problems faced.

1. The development of clinically active 5-LO inhibitors was not successful in the past and there is only one 5-LO inhibitor on the market.

2. Inhibitors of the LT pathway lacked clinical efficacy in the past or even exacerbate the disease.

We have recently published a study showing that 5-LO can also influence tumor cell function in a non-canonical way (Weisser et al. PMID 36114329). Furthermore, pharmacological inhibition of LT formation in the tumor is accompanied by a concurrent inhibition of LTs in leukocytes which interferes with the patients' anti-tumor response. Both aspects might explain the problems faced in clinical studies.

We think that the answer to these problem might be the development of inhibitors that interfere with non-canonical functions of 5-LO but spare LT formation especially in leukocytes and thus the anti-tumor response of the immune system. Furthermore, further research should concentrate on fatty acid uptake, allocation and LT release in oxylipin-dependent cancer cells and compare this to leukocytes to see if differences exist that might be targeted to prevent the formation of oxylipins and their release from cancer cells. In addition, studies on the details of the cell cycle-dependent 5-LO transcriptional regulation as well as further investigation of the interplay between 5-LO-expressing tumors and cells of the tumor stroma might lead to the identification of possible targets.

We discuss these issues now in depth in the manuscript (lines 511-539).

4. The authors show that the upregulation of 5-LO was inversely correlated with cell proliferation and activation of PI3K/mTORC-2- and MEK-1/ERK-dependent pathways. Was this response associated with the upregulation of the cyclooxygenase pathway? The authors can comment on this possibility in the discussion.

We agree with reviewer number 1 that formation of prostaglandins and regulation of cyclooxygenase enzymes along with the 5-LO pathway is an important aspect. Especially since a shunting of arachidonic acid is possible between the two pathways. Fortunately, we already had the data on PGs released alongside the LTs in our incubations since both pathways are routinely assessed in our LC/MS-MS method. There was no PG release into the cell culture supernatants during monolayer and spheroid growth as well as during Dactolisib and Cobimetinib treatment of the cells, even when arachidonic acid was supplemented. Nevertheless, we have now reinvestigated our samples for the expression of COX isoenzyme mRNA and protein. There was a profound regulation on the mRNA level but this did not translate into protein in both cell lines. We have added these findings to the results section (lines 136-147) and created a supplementary figure depicting these data (Suppl. Fig. 2).

5. The authors can comment on the impact of 5-LO inhibitors and receptor antagonists on cancer development in light of their findings.

We answered this question in our answer to question number 3.

Reviewer 2:

This article is of potentially great interest but first some basic questions need to be addressed. The findings in this report should be divided in several separate papers.

Major points: It is still very unclear whether cells that seem to contain a silent "5-LOX", really contain a full-length enzyme. The first point should be to isolate and sequence the enzyme from one of these cell lines. The authors have meritoriously found a way to increase the expression of the enzyme.

We agree with reviewer No 2 that it is vital to confirm that the 5-LO expressed in HT-29 and HCT-116 cells is the full length enzyme, capable of producing leukotrienes. In fact, in advance to the present study we asked this question ourselves and conducted the following experiments:

1. **Western Blotting:** To make sure that the band detected is 5-LO, we compared this to lysates from the same cell lines carrying a 5-LO KO. Here no band was detectable. Furthermore, the band detected and the recombinant control protein run on the same height showing that no major deletions are present.
2. **Activity assay:** We performed activity assays showing that the enzyme is active and activity is elevated in cell homogenates.

Furthermore, LTB4 is found in the incubations which shows that the 5-HETE measured is not only an autooxidation product of ARA (PMID 36114329).

3. **Sequencing of the protein:** Unfortunately, MALDI-MS sequence coverage of the 5-LO protein is poor in our hands.
4. **mRNA/cDNA sequencing:** Instead, we reverse transcribed cellular mRNA and sequenced the mature *ALOX5* transcript. This experiment showed that both cell lines express the wild-type *ALOX5* mRNA that allows the expression of a full length, non-mutated enzyme.

Due to these experiments we are confident that the 5-LO expressed in both cell lines is the wild-type enzyme.

Second major point: It is also very unclear which role 5-LOX plays in cancer. The authors must be very careful when they state that 5-LOX plays a role in cancer. Today we only know that certain B-cell lymphoma have high expression of 5-LOX; other single publications about 5-LOX and cancer have not yet been confirmed by other independent researchers. Furthermore, there seems to be a substantially auto-oxidative formation of 5-HETE in several experiments.

We agree with reviewer number 2 that a number of publications dealing with the role of 5-LO in cancer suffer from technical problems such as the non-adequate handling of samples leading to a high auto-oxidative background or unspecific antibodies. Furthermore, many studies are based on pharmacologic data only, using inhibitors that display a number of pleiotropic effects such as interference with prostaglandin formation/release thus making correct data interpretation impossible. We have thoroughly validated our antibodies. In activity assays cell free controls are included to rule out excessive autooxidation.

Concerning the role of 5-LO in cancer: There is substantial body of evidence in literature proving the overexpression of 5-LO in many different cancers. A number of RNAi experiments has confirmed a role of the enzyme in cell proliferation and survival. Furthermore, we could recently show that the enzyme influences spheroid formation, migration and proliferation of a number of overexpressing cell lines employing a complete KO (Weisser et al, Cancer Gene Ther., 2022; PMID 36114329).

Reviewer 3:

In this article, Gobel et al. showed how lipoxygenase-derived lipid mediator biosynthesis is also affected by 3D growth. They have studied the 5-LOX regulation and effects on several pathways in experiments using both 2D and 3D cultures of two colon cancer cell lines. A number of in vitro experiments were conducted, and the data is well presented. However, this reviewer does not see the novelty in the research. All the studies performed are in vitro (Spheroids).

We disagree here with Reviewer 3. It is highly relevant to study 5-LO and lipid mediator formation in 3D cell culture to better understand what regulates eicosanoid formation in human solid tumors. Furthermore, the relation between cell cycle arrest / quiescence and 5-LO expression has never been shown before.

Moreover, the manuscript is too lengthy.

We have now considerably shortened the introductory part as well as the discussion section to avoid lengthiness of the manuscript.

We would like to thank the reviewers for their comments. Their input considerably improved the manuscript. Therefore, we hope that the revised version may now be found acceptable for publication in *Life Science Alliance*.

Thank you very much for considering our revised manuscript. We look forward to hearing from you.

February 8, 2023

RE: Life Science Alliance Manuscript #LSA-2022-01804-TR

Dr. Astrid S Kahnt
Goethe University Frankfurt
Institute of Pharmaceutical Chemistry
Max-von-Laue-Straße 9
Frankfurt 60438
Germany

Dear Dr. Kahnt,

Thank you for submitting your revised manuscript entitled "Three-dimensional growth reveals fine-tuning of 5-lipoxygenase by proliferative pathways in cancer". We would be happy to publish your paper in Life Science Alliance pending final revisions necessary to meet our formatting guidelines.

- please upload your supplementary figures as single files
- please add the Twitter handle of your host institute/organization as well as your own or/and one of the authors in our system
- please make sure that every author that is listed in the manuscript is also listed in our system and that the author order in both the manuscript and the system match
- you may want to consider submitting Figure 10 as a Graphical Abstract rather than as a figure, up to you

A. FINAL FILES:

B. MANUSCRIPT ORGANIZATION AND FORMATTING:

**Submission of a paper that does not conform to Life Science Alliance guidelines will delay the acceptance of your

manuscript.**

The license to publish form must be signed before your manuscript can be sent to production. A link to the electronic license to publish form will be sent to the corresponding author only. Please take a moment to check your funder requirements.

Sincerely,

Reviewer #1 (Comments to the Authors (Required)):

This is an interesting and well-performed study. Many experiments were carried out utilizing appropriate techniques. The revised version of the manuscript is improved, and they addressed most of the points raised by the authors and replied appropriately to reviewers' comments.

February 16, 2023

RE: Life Science Alliance Manuscript #LSA-2022-01804-TRR

Dr. Astrid S Kahnt
Goethe University Frankfurt
Institute of Pharmaceutical Chemistry
Max-von-Laue-Straße 9
Frankfurt 60438
Germany

Dear Dr. Kahnt,

Thank you for submitting your Research Article entitled "Three-dimensional growth reveals fine-tuning of 5-lipoxygenase by proliferative pathways in cancer.". It is a pleasure to let you know that your manuscript is now accepted for publication in Life Science Alliance. Congratulations on this interesting work.

DISTRIBUTION OF MATERIALS:

Again, congratulations on a very nice paper. I hope you found the review process to be constructive and are pleased with how the manuscript was handled editorially. We look forward to future exciting submissions from your lab.

Sincerely,
